# AlphaMath Almost Zero: Process Supervision without Process

**Guoxin Chen**[*], **Minpeng Liao**[*], **Chengxi Li**[*], **Kai Fan**[*][†]
Tongyi Lab
chenguoxin22@mails.ucas.ac.cn
{minpeng.lmp,xiji.lcx,k.fan}@alibaba-inc.com
Code: https://github.com/MARIO-Math-Reasoning/Super_MARIO

## Abstract

Although recent advancements in large language models (LLMs) have significantly improved their performance on various tasks, they still face challenges with complex and symbolic multi-step reasoning, particularly in mathematical reasoning. To bolster the mathematical reasoning capabilities of LLMs, most existing efforts concentrate on seeking assistance from either domain experts or GPT-4 for high-quality process-supervised data, which is not only expensive but also labor-intensive. In our study, we propose an innovative framework, AlphaMath, that bypasses the need for process annotations (from humans or GPTs) by leveraging Monte Carlo Tree Search (MCTS). This framework focuses on unleashing the potential of a well-pretrained LLM to autonomously enhance its mathematical reasoning. Specifically, we integrate a value model with the LLM, automatically generating both process supervision and step-level evaluation signals in MCTS. Furthermore, we propose an efficient inference strategy—step-level beam search, where the value model is crafted to assist the policy model (*i.e.*, LLM) in navigating more effective reasoning paths, rather than solely relying on prior probabilities. The experimental results on both in-domain and out-of-domain datasets demonstrate that even without GPT-4 or human-annotated process supervision, our AlphaMath framework achieves comparable or superior results to previous state-of-the-art methods.

## 1 Introduction

Recent studies have extensively explored how to improve mathematical reasoning in large language models (LLMs) [27, 1, 37, 34, 2, 35]. An effective approach [46, 38, 14, 21, 31, 25] is to artificially inject external knowledge into LLMs through fine-tuning on a substantial volume of high-quality, process-supervised data (*i.e.*, solutions). As shown in Table 1, the annotation of high-quality solutions in current efforts primarily relies on domain experts or GPT-4 [27]. However, due to trillions of training tokens and billions of parameters, existing LLMs possess a vast reservoir of knowledge, which remains underutilized in current finetuning-based approaches.

Table 1: Annotation Cost

| Annotation Source | | Methods |
| --- | --- | --- |
| Human | GPT-4 | |
| ✓ | ✓ | [21, 46] |
| ✗ | ✓ | [14, 25, 38] |
| ✗ | ✗ | Ours |

To more effectively harness the intrinsic knowledge of LLMs, advanced prompting techniques, such as Program-of-Thought (PoT) [6] and Program-Aided Language (PAL) [13], have been developed, integrating the in-context learning proficiency with external tools such as code interpreter to handle precise numerical and symbolic computation. However, these approaches have not fully unleashed

---

[*]equal contribution
[†]Corresponding Author.

38th Conference on Neural Information Processing Systems (NeurIPS 2024).

the potential of LLMs and often rely on self-consistent majority voting [39], which does not reflect the natural process by which humans solve mathematical problems. This discrepancy arises because both the PoT and PAL frameworks pursue a solution to its final answer regardless of the accuracy of intermediate steps. Unlike these approaches, humans tend to reassess and potentially alter their solution path upon encountering a mistake or dead-end in the problem-solving process. In this manner, humans iteratively enhance their self-cognition and reinforce the utilization of knowledge.

In this research, we aspire for LLMs to possess the similar ability as humans to realize self-evolution and strengthen their utilization of knowledge autonomously. Notably, AlphaGo Zero [33] showcases how a neural network model can progressively evolve without human knowledge, autonomously producing the Go game training strategies. For the strategy (*i.e.*, solution) of mathematical problems, both textual analysis [45] and code snippets [14] demand rigorous logical structuring. Consequently, most finetuning-based approaches concentrate on seeking assistance from domain experts or GPT-4 for annotated solutions, thereby overlooking the reservoir of knowledge inherent in LLMs.

Instead, we hypothesize that well pre-trained LLMs already possess the necessary mathematical knowledge to generate correct reasoning; however, they require appropriate stimulation—such as an improved prompt or search strategy—to do so. In this work, solutions including both textual analysis and code snippets are autonomously generated by a well pre-trained LLM equipped with appropriate prompts and deliberately designed Monte Carlo Tree Search (MCTS) framework [4, 32]. Specifically, we integrate LLMs with the MCTS to strike a more effective balance between exploration and exploitation, enabling the generation of high-quality process-supervised solutions without professional human annotations. To enhance the efficiency of solution generation, we incorporate a value model into the same LLM by appending a linear layer. This advancement removes the necessity for time-consuming rollouts for reward estimation. While the LLM learns to solve mathematical problems from its own annotated solutions, the value model simultaneously learns how to assess the quality of intermediate reasoning steps from the corresponding state values in MCTS, just like humans.

During the inference stage, with the value model, LLMs can perform MCTS inference, which significantly enhances their reasoning capabilities but limited by efficiency. Therefore, inspired by beam search algorithm [36], we propose a step-level beam search strategy, where the value model is crafted to assist the policy model (*i.e.*, LLM) in navigating more effective solution paths, as opposed to relying solely on prior probabilities. Compared to the greedy or MCTS inference strategies, the step-level beam search significantly enhances the LLM's reasoning capability at a minimal cost.

Empirically, we build an iterative training framework as shown in Figure 1. Unlike in the game of Go, where the final board state directly indicates a win or loss, our methodology requires validation of the equivalence between predicted answers and actual ones. This is the fundamental reason why our training data necessarily consists of question statements and their final answers. Furthermore, we validate the applicability of our framework on three popular types of LLMs: domain-specific pre-trained models [31], general-purpose pre-trained models [11], and supervised fine-tuned models [21]. Our contributions are summarized as follows:

- We propose a novel approach that integrates a pre-trained LLM with a deliberately designed Monte Carlo Tree Search (MCTS) framework. This combination allows LLMs to autonomously generate high-quality mathematical reasoning solutions without the need for professional human annotations, leading to a more efficient utilization of their inherent knowledge.
- To address the efficiency limitations of MCTS inference, we propose a step-level beam search strategy, which introduces a lightweight value model that works alongside the LLM, enabling the simultaneous assessment of the quality of intermediate reasoning steps. This method parallels human problem-solving by allowing the LLM to learn from its own solutions while also evaluating the effectiveness of its reasoning strategy, thus enhancing the overall reasoning capabilities.
- Extensive experiments demonstrate that our AlphaMath can effectively stimulate the internal knowledge of LLMs, achieving better or on par task performance on both in-domain and out-of-domain mathematical reasoning datasets, even without any process annotations.

## 2  Preliminary

We assume that, for any given input question $\mathbf{q}$, the solution process can be broken into multiple reasoning steps (*e.g.*, segmenting the solution based on distinct stages or simply on a period). From this perspective, we conceptualize mathematical problem solving within the context of reinforcement

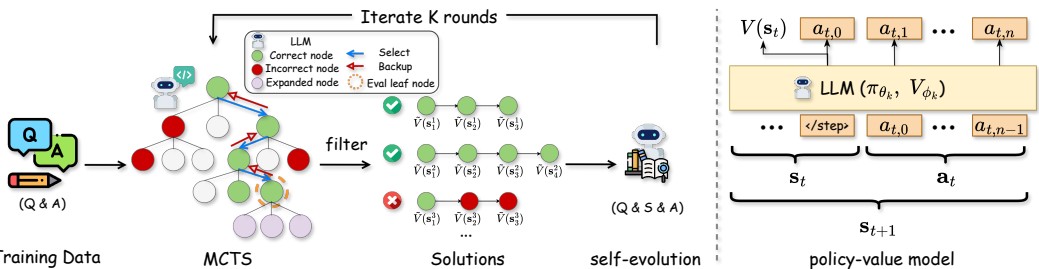

Figure 1: Our approach involves iterating through three distinct stages. (1) Collect a mathematical dataset that comprises questions and their corresponding final answers. (2) Employ MCTS on the policy and the value model to generate both correct and incorrect solution paths along with state values. (3) Optimize the policy and the value model with generated data from MCTS.

learning. Concretely, consider a complete solution consisting of $T$ reasoning steps. At a given time $t$, we represent the partial solution as the state $\mathbf{s}_t$, and the subsequent reasoning step that might be taken as the action as $\mathbf{a}_t$. For detailed definitions and examples of our reasoning step, please refer to Appendix C.1. In this scenario, the policy model is embodied by a large language model, and the transition $f(\mathbf{s}_{t+1}|\mathbf{a}_t, \mathbf{s}_t)$ from one state to the next is deterministically accomplished through the concatenation operation.

$$\pi_\theta(\mathbf{a}_t|\mathbf{s}_t) = \text{LLM}(\mathbf{a}_t|\mathbf{s}_t), \qquad \mathbf{s}_{t+1} = \text{Cat}(\mathbf{s_t}, \mathbf{a}_t) \tag{1}$$

Our primary goal is to develop a step-level value model, denoted as $V_\phi(\mathbf{s})$, which is capable of assessing the expected returns from the current partial solution and guiding the LLM to select more reasonable subsequent reasoning steps.

To train the value model, we first define the reward in the context of mathematical problem solving, by assigning the reward $r = 0$ to all non-terminal reasoning steps, and $r = \pm 1$ to a correct/incorrect final answer. A common method to create the training signal is to employ Monte Carlo (MC) evaluation $\widetilde{V}(\mathbf{s}_t) = \frac{1}{N}\sum_{i=1}^{N} r\left(\mathbf{a}_{t'\geq t}^{(i)}, \mathbf{s}_{t'>t}^{(i)}|\mathbf{s}_t\right)$, where $\mathbf{a}_{t'\geq t}^{(i)}$ and $\mathbf{s}_{t'>t}^{(i)}$ represent the actions and states in the $i$-th simulation sampled by the policy model and the state transition function. $r(\cdot|\mathbf{s}_t)$ means the reward of the final outcome in one simulation from state $\mathbf{s}_t$. Then, for any given partial solution $\mathbf{s}$, we can train the step-level value model $V_\phi$ using a regression loss defined as follows:

$$\mathcal{L}_{V_\phi}(\mathbf{s}) = \left\|V_\phi(\mathbf{s}) - \widetilde{V}(\mathbf{s})\right\|^2. \tag{2}$$

## 3 AlphaMath

In the above approach of MC evaluation, it requires multiple simulations from each state, which may be inefficient in practice. We propose employing the Monte Carlo Tree Search (MCTS) algorithm, which has the potential to reuse simulations and update the estimated values in a principled manner.

### 3.1 MCTS Evaluation

As shown in Figure 1, our approach employs iterative training. Before the $(k+1)$-th round training, we have a value model $V_{\phi_k}$ and a LLM policy model $\pi_{\theta_k}$, which are the same model but with different final layers in our paper. Using these models, we can construct an inference algorithm powered by MCTS. This algorithm starts with the initial state as its root, and through the synergistic use of the policy and value models, systematically grows the search tree by adding new nodes. These nodes correspond to the states deemed to have high potential based on the outcomes of simulated trajectories. Specifically within the context of mathematical problem-solving, as shown in Figure 2, we customize the four key operations of the MCTS algorithm as follows:

**Selection** During the $i$-th simulation of the MCTS, the process begins with $\mathbf{s}_0$, representing the initial state containing the input question. The algorithm then proceeds to explore the tree $\mathcal{T}_k$ by selecting

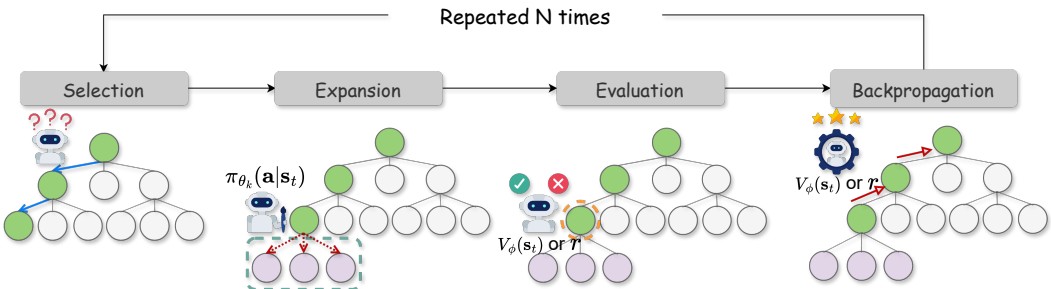

Figure 2: An overview of the four key operations in MCTS

nodes according to a variant of the PUCT algorithm [30]. This selection process is mathematically represented as:

$$\mathbf{a}_t = \arg\max_{\mathbf{a}\in\mathcal{T}_k}\left[\hat{Q}(\mathbf{s}_t,\mathbf{a}) + c_{\text{puct}}\pi_{\theta_k}(\mathbf{a}|\mathbf{s}_t)\frac{\sqrt{N_{parent}(\mathbf{a})}}{1+N(\mathbf{s}_t,\mathbf{a})}\right] \tag{3}$$

where the state-action value $\hat{Q}(\mathbf{s},\mathbf{a})$ and its visiting count $N(\mathbf{s},\mathbf{a})$ are stored in the tree and will be updated as the search progresses. $N_{parent}(\mathbf{a})$ represents the visiting count of the parent node of $\mathbf{a}$. The action selection iterates until it encounters a leaf node of the current search tree. In our case, the prior $\pi(\mathbf{a}|\mathbf{s}_t)$ is defined as the exponential of averaged log-probability of all tokens in the step $\mathbf{a}$, *i.e.*, $\exp\left(\frac{1}{|\mathbf{a}|}\sum\log\pi(a_j|\mathbf{a}_{<j},\mathbf{s}_t)\right)$.

**Expansion** Back-tracing from the selected leaf node to the root forms a partial solution, serving as a prompt for further node expansions. In our case, given that the LLM can theoretically generate an unlimited number of potential actions (token sequence), we employ sampling generation with higher temperature to ensure diversity.

**Evaluation** Evaluation of the leaf node or partial solution $\mathbf{s}_t$, identified after the selection phase, is conducted by weighted sum as introduced in [32, 33].

$$\hat{V}(\mathbf{s}_t)^{(i)} = (1-\lambda)\cdot V_{\phi_k}(\mathbf{s}_t) + \lambda\cdot r\left(\mathbf{a}^{(i)}_{t'\geq t},\mathbf{s}^{(i)}_{t'>t}|\mathbf{s}_t\right) \tag{4}$$

The intermediate value estimation $\hat{V}$ in MCTS differs from the training signal $\widetilde{V}$ defined in preliminary section 2. The parameter $\lambda$ serves to balance the contribution of the value model's estimation with the empirical reward obtained during the rollout.

In our case, we follow a trade-off rollout strategy between AlphaGo [32] and AlphaGo Zero [33]. Because our tree depth is much shallower than Go games (*e.g.*, a maximum depth of 8) and expansions can easily reach a terminal node, we set an indicator function $\lambda = \mathbb{I}_{\text{terminal}}(\mathbf{s}_t)$. If the expanded node is terminal, the reward is returned; otherwise, the value is predicted by the model $V_{\phi_k}$.

**Backup** We did not make any modifications to the backup. At the end of the $i$-th simulation, each edge $(\mathbf{s},\mathbf{a})$ along the path from the leaf node $\mathbf{s}_t$ to the root undergoes a backward pass update. The updates to their state-action values and visiting counts are executed according to the following rules: $N(\mathbf{s},\mathbf{a}) \leftarrow N(\mathbf{s},\mathbf{a}) + 1$ and $\hat{Q}(\mathbf{s},\mathbf{a}) \leftarrow \frac{1}{N(\mathbf{s},\mathbf{a})}\sum_{j=1}^{i}\mathbb{I}_{\mathbf{s},\mathbf{a}\to\mathbf{s}_t}\hat{V}(\mathbf{s}_t)^{(j)}$.

**Value Estimation** After running $N$ simulations with the MCTS algorithm, we obtain the final tree $\mathcal{T}_k$, which stores the expanded nodes and their corresponding state-action values $Q(\mathbf{s},\mathbf{a})$. Considering that the transition function is deterministic, and assuming that $Q(\mathbf{s}_t,\mathbf{a}_t) = r(\mathbf{s}_t,\mathbf{a}_t) + V(\mathbf{s}_{t+1}) = V(\mathbf{s}_{t+1})$ for non-terminal nodes[3], we can employ the $Q$ values as training signals. This implies that we can directly fit the state-action value of non-terminal nodes as,

$$\widetilde{V}(\mathbf{s}_{t+1}) = \hat{Q}(\mathbf{s}_t,\mathbf{a}_t) \tag{5}$$

---

[3]Reward is 0 for non-terminal node, and reward is determined by the final answer in terminal node.

---

**Algorithm 1** Inference with MCTS

---

**Require:** $B_1 = 1$, question $\mathbf{q}\,(\mathbf{s}_0)$, policy / value models $\pi_\theta, V_\phi$, simulations $N$, max depth $T$.

1:   Build the complete tree $\mathcal{T}$ by running $\text{MCTS}_{\pi_\theta, V_\phi}(\mathbf{s}_0, N, T)$.

2:   $\mathcal{C} = [\mathbf{s}_0], t = 0$                                                    ▷ *Initialize candidates*

3:   **while** $t < T$ **and** non-terminal path in $\mathcal{C}$ **do**

4:       Initialize priority queue $\mathcal{C}_{t+1}$                              ▷ *Max heap*

5:       **for** $\mathbf{s}_t$ in $\mathcal{C}$ **do**

6:           **for** $\mathbf{a}$ in $\mathcal{T}_{\text{children}}(\mathbf{s}_t)$ **do**               ▷ *Directly get children from tree*

7:              $\mathbf{s_{t+1}} = \text{Cat}\,[\mathbf{s}_t, \mathbf{a}]$

8:              Add $(\mathbf{s_{t+1}}, \mathcal{T}_Q(\mathbf{s}_t, \mathbf{a}))$ to $\mathcal{C}_{t+1}$      ▷ *Directly get Q-value from tree*

9:       $\mathcal{C} \leftarrow$ Top-$B_1$ of $\mathcal{C}_{t+1}$

     **return** Top-1 of $\mathcal{C}$                                  ▷ *Return top-1 as the final solution path*

---

---

**Algorithm 2** Step-level Beam Search

---

**Require:** Beam sizes $B_1$, $B_2$, question $\mathbf{q}\,(\mathbf{s}_0)$, policy / value models $\pi_\theta, V_\phi$, max steps $T$.

1:   $\mathcal{C} = [\mathbf{s}_0] * B_1, t = 0$                                      ▷ *Initialize candidates*

2:   **while** $t < T$ **and** non-terminal path in $\mathcal{C}$ **do**

3:       Initialize priority queue $\mathcal{C}_{t+1}$                              ▷ *Max heap*

4:       **for** $\mathbf{s}_t$ in $\mathcal{C}$ **do**

5:          Sample $\{\mathbf{a}^{(b)}\}_{b=1}^{B_2} \sim \pi_\theta(\mathbf{a}|\mathbf{s}_t)$         ▷ *LLM generates $B_2$ samples in parallel.*

6:          **for** $b = 1$ to $B_2$ **do**

7:              $\mathbf{s_{t+1}} = \text{Cat}\,[\mathbf{s}_t, \mathbf{a}^{(b)}]$

8:              Add $(\mathbf{s_{t+1}}, V_\phi(\mathbf{s_{t+1}}))$ to $\mathcal{C}_{t+1}$        ▷ *$V_\phi(\mathbf{s_{t+1}})$ predicted by value model*

9:       $\mathcal{C} \leftarrow$ Top-$B_1$ of $\mathcal{C}_{t+1}$

     **return** Top-1 of $\mathcal{C}$                                  ▷ *Return top-1 as the final solution path*

---

## 3.2   Iterative Training

**Initialization** Initially, our approach begins with a pre-trained LLM as the policy model $\pi_{\theta_1}$. We extend this model by adding an auxiliary linear layer with a `tanh` activation function, which works alongside the traditional softmax layer responsible for token prediction, as depicted in the rightmost panel of Figure 1. This design implies that these two models, $\pi_\theta$ and $V_\phi$, share the majority of their parameters. The parameters of the linear layer associated with $V_{\phi_1}$ are randomly initialized, leading to an initial tendency of the value head to predict a value close to 0 at the first ($k = 1$) round of MCTS. However, as the simulations in the first round MCTS proceed, the rewards ($\pm 1$) from terminal nodes are back-propagated to their parent nodes. As simulations $N$ gradually increase, the estimated values $\hat{Q}$ of intermediate nodes converge towards the underlying true value within the range of $[-1, 1]$.

**Training Method** From the tree $\mathcal{T}_k$ constructed from the $k$-th round of MCTS, we can sample solution paths corresponding to terminal nodes with correct and incorrect predicted answers, denoted as $\mathbf{x}^+$ and $\mathbf{x}^-$, respectively, together with the value estimation of each node along these paths. We then apply a multi-task loss function to update both the policy and value models.

$$\arg\min_{\theta,\phi} -\log \pi_\theta(\mathbf{x}^+|\mathbf{q}) + \beta \cdot \left( \sum_{t=1}^{T(\mathbf{x}^+)} \|V_\phi(\mathbf{s}_t) - \widetilde{V}(\mathbf{s}_t)\|^2 + \sum_{t=1}^{T(\mathbf{x}^-)} \|V_\phi(\mathbf{s}_t) - \widetilde{V}(\mathbf{s}_t)\|^2 \right) \quad (6)$$

where the first term represents the negative log-likelihood loss for next-token prediction in correct solutions, and the second term within the big brackets captures the loss in value prediction for both correct and incorrect solutions, respectively. $T(\mathbf{x})$ denotes the number of steps for solution path $\mathbf{x}$. $\beta$ is a tunable hyper-parameter to control the weight of value loss. With the updated policy and value models $\pi_{\theta_{k+1}}$ and $V_{\phi_{k+1}}$, we can advance to the next-round MCTS, iterating this training process to enhance our models further.

## 3.3   Inference

**MCTS** For MCTS inference, it is necessary to set $\lambda = 0$ in the evaluation of Eq. (4). Unlike in board games, we cannot verify the correctness of a path during inference. Therefore, we consistently

rely on the value model for node evaluation, including for terminal nodes. MCTS demands multiple simulations to update visiting counts and $Q$ values, aiming to estimate a robust policy distribution.

After the tree has been completely built, the algorithm iteratively selects the top-$B_1$ steps (usually $B_1 = 1$ in MCTS) from the root in a top-down manner. This selection is guided by the *maximum Q-value* stored in the child nodes of the tree. Subsequently, all child nodes from the previously selected $B_1$ steps are collectively re-ranked based on their $Q$-values, and the top-$B_1$ nodes from this ranking are retained for the next iteration. A summary of the algorithm can be found in Algorithm 1.

**Step-level Beam Search** However, MCTS is computationally intensive for simulations, making it less viable for use in production environments. To address this, we modify the MCTS inference process by eliminating the backup operation, introducing a simplified method, which we refer to as Step-level Beam Search (SBS), detailed in Algorithm 2. This approach does not construct the entire tree; instead, it dynamically selects the best child node during node expansion.

There are two primary technical distinctions in SBS. First, since node expansion is required on the fly, we introduce a new beam size, $B_2$, to represent the maximum number of node expansions. Second, the selection criterion no longer relies on the $Q$-value converged after $N$ simulations but instead uses the *maximum value prediction* directly from the value model. Importantly, with special case SBS $B_1 = 1$ as a fast approximation of MCTS, it facilitates the sequential, streaming output of steps, rendering it more suitable for practical implementation in real-world production.

## 4 Experiments

### 4.1 Experimental Setup

In this study, we mainly investigate the math domain-specific language model, DeepSeekMath-Base-7B [31], pre-trained on a substantial math-related corpus without any supervised fine-tuning (SFT), which is believed to possess necessary mathematical knowledge to tackle a wide range of mathematical problems.

**Training Data Generation via MCTS** For the training sets, we exclusively extract question and answer pairs from GSM8K [7] and MATH [15], omitting the human-annotated solution analysis. *In total, our training set includes only 15k question-answer pairs and 0 solution process.*

In our setup, we utilize the MCTS framework to generate detailed solution processes equipped with the Python code interpreter. Initially, for the first round of MCTS, the prompt used for our solution generation adheres to the REACT [43] format, incorporating 2 demonstrations randomly selected from a pool of 20 prepared examples. Starting from the second round, with an already fine-tuned model from the first round, we employ a straightforward prompt in our SFT XML format without any demonstration. Two prompt examples are shown in Appendix F.

Specifically, we iteratively generate data and train our policy and value models through $K = 3$ rounds, continuing until the enhancement observed between any two consecutive rounds is incremental. In every round, we build 10 trees for each question-answer pair and randomly sample at most 4 correct and 4 incorrect solution processes. The ratio between positive and negative examples is approximately 1:1, with the count of positive examples in each round varying between 57k and 59k.

**Test Data** We evaluate our approach not only on GSM8K and MATH but also on out-of-distribution (OOD) datasets GaoKao2023 [21] and OCWCourses [19]. These two OOD datasets are even more challenging than MATH. Please refer to Appendix C.5 for more details about the dataset statistics. To assess the accuracy of the predicted answers, we utilize the math evaluation toolkit [47].

**Baselines** We first compare our approach with strong proprietary and open-source models, including OpenAI's ChatGPT and GPT-4 [27], Llama2 [37], Llemma [3]. By default, we report the results obtained using Chain of Thought (CoT) prompting [40], along with the prompting results of PAL [13], due to its enhanced performance in mathematical reasoning.

SFT models leverage high-quality seed data with process supervision derived from GPT-4 or humans to enhance their reasoning capabilities. To ensure a fair comparison, we primarily contrast our approach with the highest-performing SFT models that utilize an external tool - a Python code interpreter. These include MAmmoTH [46], MathCoder [38], ToRA [14], MARIO [21], MathGenie [25], and DeepSeek-Math-Instruct [31]. More implementation details can be found in Appendix C.

Table 2: Main results. The best results of open-sourced models are bold. For the methods with released model's outputs, performance metrics using the evaluation toolkit [47] are also provided in brackets. ‡Seed data refers to high-quality annotated (question, solution) pairs, typically annotated by humans or GPT-4. §Unless otherwise specified, we set beam size $B_2 = 5$ in SBS and number of simulations $N = 40$ in MCTS by default.

| Model | Size | Seed Data‡ Annotation | Seed Data Size | Tool | Zero Shot | In-Domain GSM8K | In-Domain MATH | OOD GK2023 | OOD OCW |
|---|---|---|---|---|---|---|---|---|---|
| *Proprietary Models* | | | | | | | | | |
| GPT-4 | - | - | - | ✗ | ✗ | 92.0 | 42.5 | - | - |
| GPT-4 (PAL) | - | - | - | ✓ | ✗ | 94.2 | 69.7 | 43.6 | 30.1 |
| ChatGPT | - | - | - | ✗ | ✗ | 80.8 | 35.5 | - | - |
| ChatGPT (PAL) | - | - | - | ✓ | ✗ | 78.6 | 38.7 | - | - |
| Gemini-1.5 Pro | - | - | - | ✗ | ✗ | 91.7 | 58.5 | - | - |
| Claude-3.5-Sonnet | - | - | - | ✗ | ✓ | 96.4 | 71.1 | - | - |
| *Open-Source Models* | | | | | | | | | |
| Llama-2 | 7B | - | - | ✗ | ✗ | 13.3 | 4.1 | - | 3.7 |
| CodeLlama | 7B | - | - | ✗ | ✗ | 10.5 | 4.5 | - | 4.7 |
| CodeLlama(PAL) | 7B | - | - | ✓ | ✗ | 27.1 | 17.2 | - | - |
| Llemma | 7B | - | - | ✗ | ✗ | 36.4 | 18.0 | - | 7.7 |
| Llemma(PAL) | 7B | - | - | ✓ | ✗ | 40.1 | 21.5 | - | - |
| DeepSeekMath-Base(PAL) | 7B | - | - | ✓ | ✗ | 66.9 | 31.4(33.2) | - | - |
| *SFT Models* | | | | | | | | | |
| MAmmoTH-Coder | 34B | GPT-4+Human | 260k | ✓ | ✓ | 72.7 | 43.6 | 25.2 | 14.0 |
| MathCoder | 34B | GPT-4 | 49k | ✓ | ✓ | 81.7 | 46.1(45.8) | - | - |
| ToRA-Code | 34B | GPT-4 | 16k | ✓ | ✓ | 80.7 | 50.8(51.2) | 31.7 | 5.5 |
| MARIO | 34B | GPT-4+Human | 27k | ✓ | ✓ | 78.2 | 53.5 | 42.6 | 30.2 |
| MathGenie | 34B | GPT-4 | 80k | ✓ | ✓ | **84.1** | 55.1 | - | - |
| Llama-2 SFT | 7B | Human | 15k | ✗ | ✓ | 41.3 | 7.2 | - | - |
| Llama-2 RFT | 7B | Human | 15k | ✗ | ✓ | 51.2 | - | - | - |
| MAmmoTH-Coder | 7B | GPT-4+Human | 260k | ✓ | ✓ | 59.4 | 33.4 | 15.3 | 11.0 |
| MathCoder | 7B | GPT-4 | 49k | ✓ | ✓ | 67.8 | 30.7(30.6) | - | - |
| ToRA | 7B | GPT-4 | 16k | ✓ | ✓ | 68.8 | 40.1 | 19.5 | 2.6 |
| ToRA-Code | 7B | GPT-4 | 16k | ✓ | ✓ | 72.6 | 44.6 | 23.9 | 4.8 |
| MARIO | 7B | GPT-4+Human | 27k | ✓ | ✓ | 74.5 | 48.3 | 34.5 | 21.7 |
| MathGenie | 7B | GPT-4 | 80k | ✓ | ✓ | 76.0 | 48.3 | - | - |
| DeepSeekMath-Instruct | 7B | GPT-4+Human | 776k | ✓ | ✓ | 83.7 | 57.4(57.2) | 43.9 | 18.0 |
| DeepSeekMath-Base | 7B | | | | | | | | |
|   +our prompt 2-shot | | - | - | ✓ | ✗ | 59.7 | 33.2 | 21.9 | 9.2 |
|   +AlphaMath ($K = 3$) | | ✗ | 0 | ✓ | ✓ | 73.5 | 53.6 | 40.5 | 26.1 |
|    + SBS§ ($B_1 = 1$) | | ✗ | 0 | ✓ | ✓ | 81.1 | 62.8 | 46.2 | 30.5 |
|    + SBS ($B_1 = 3$) | | ✗ | 0 | ✓ | ✓ | **84.1** | **66.3** | **51.4** | 33.1 |
|    + MCTS ($B_1 = 1$) | | ✗ | 0 | ✓ | ✓ | 83.2 | 64.0 | 48.4 | **33.8** |

## 4.2 Main Results

We report our in-domain and out-of-domain (OOD) results in Table 2. Different from previous works [46, 38, 14, 21, 25], our proposed AlphaMath does not rely on high-quality solutions annotated by humans or GPT-4, whether in the form of text analysis or code snippets. Such solutions typically bolster the model's reasoning abilities but also entail substantial costs associated with annotation. Furthermore, our method differs from prior research by not incorporating any external datasets (*e.g.*, new questions and solutions) beyond the GSM8K and MATH datasets. The last five rows of Table 2 present our principal findings.

**First**, we establish a baseline with the inherent mathematical reasoning ability of DeepSeekMath-Base using our designed prompt in a 2-shot setting. It's important to note that this outcome differs from the results reported for DeepSeekMath-Base (PAL) in the original study, as it utilized prompts with 8-shot and 4-shot for the GSM8K and MATH datasets, respectively. **Secondly**, we only evaluate the policy model with greedy decoding. In comparison to our initial study, we record an enhancement of about 20 points for challenging problems in the MATH, GaoKao2023 (GK2023), and OCWCourses (OCW) datasets, and an improvement of more than 10 points for grade school math problems. **Thirdly**, we delve into the role of the value model in facilitating mathematical reasoning, utilizing a computationally efficient step-level beam search (SBS) in Algorithm 2. When we increment $B_1$ with a default $B_2 = 5$ and temperature of 1.0, a corresponding gradual improvement in performance is observed. More discussion about the temperature in SBS can refer to Appendix 4.7. **Ultimately**, we evaluate our approach in Algorithm 1. In contrast to the training data generation, we construct a single tree with 40 simulations, a maximum of 5 child nodes, and a temperature of 0.6. While MCTS

demonstrates improved performance on more challenging datasets, attributed to its expansive search space, its substantial computational demands curtail its practical applicability in real-world scenarios.

In summary, our approach demonstrates that, even in the absence of high-quality GPT-4 or human-annotated solution processes, it remains competitive with or surpasses the performance of the state-of-the-art (SOTA) on 7B LLMs.

## 4.3 Analysis 1: Performance of each round

We evaluate the problem-solving rate in the MATH training dataset, which categorizes each problem by difficulty level. As shown in Figure 3, it becomes evident that MCTS achieves greater success in solving more challenging problems in subsequent rounds. In Figure 4, our findings show a general increase in performance with additional rounds of training across all strategies, applicable to both in-domain and out-of-domain test sets. Therefore, we can conclude that the quality of our self-generated training data improves incrementally with each round, and this enhancement is reflected in the performance on the test set. More analysis can refer to Appendix B.3.

## 4.4 Analysis 2: Performance of different inference strategies

We explore the performance of our model under various inference strategies including greedy decoding, step-level beam search, and MCTS. The results of MATH and GaoKao2023 are illustrated in Figure 4, while the results of other datasets can be found in Appendix B.1. Specifically, for SBS, an enhancement in performance was observed with an increase in the beam size $B_1$. MCTS exhibited the higher performance than its approximation SBS ($B_1 = 1$), but we previously noted its significant time consumption and computational inefficiency. Consequently, we provide a summary of the average problem-solving duration and the average number of intermediate steps taken on the MATH dataset in Table 3. The results indicate that MCTS demands the longest solving time and the highest number of steps, attributable to our configuration of 40 simulations. To achieve similar accuracy, step-level beam search is more computationally friendly. Additionally, we observe an intriguing phenomenon: a larger beam size $B_1$ tends to reduce the average problem-solving duration. This can be attributed to the decrease in the number of average steps required when a larger $B_1$ is employed.

**Discussion of Majority Voting** It is challenging to directly compare maj@5 with step-level beam search due to the inherent differences in their methodologies. Generally speaking, as Algorithm 2, SBS will eventually return the top-1 final answer based on the value model, while maj@5 will generate all 5 possible final answers and vote the majority for evaluation.

From the step-level perspective, maj@5 will maintain 5 candidates for the current step to generate another 5 candidates for the next step. In contrast, the SBS (*e.g.*, $B_1 = 1, B_2 = 5$) will always retain the top-1 candidate, discarding the 4 others. This provides the advantage of step-

Table 3: Analysis of Computational Efficiency on MATH dataset. # Sol. denotes the number of solutions obtained eventually.

| Inference Strategy | Acc. | Avg. time (s) | Avg. steps | # Sol. |
|---|---|---|---|---|
| Greedy | 53.62 | 1.6 | 3.10 | 1 |
| Maj@5 | 61.84 (+8.22) | 2.9 | 2.88 | 5 |
| SBS ($B_1 = 1$) | 62.80 (+9.18) | 3.1 | 3.01 | 1 |
| SBS ($B_1 = 2$) | 64.66 (+11.04) | 2.4 | 2.36 | 2 |
| SBS ($B_1 = 3$) | 66.30 (+12.68) | 2.3 | 2.21 | 3 |
| SBS ($B_1 = 5$) | 65.98 (+12.37) | 4.7 | 2.26 | 5 |
| MCTS ($B_1 = 1$) | 64.02 (+10.40) | 10.1 | 3.76 | n |

by-step streaming output in real-world production, whereas maj@5 can only output the complete solution until the voting is finalized. To sum up, their specific mechanics of candidate selection and retention differ significantly.

## 4.5 Analysis 3: Value model

In the left panel of Figure 5, we plot the fitted distribution of $Q$-values (as defined in Eq. (5)) on MATH training set for intermediate steps. For correct solutions, the distribution is markedly skewed towards a value of 1. In contrast, the distribution for incorrect solutions exhibits a lower degree of skewness, albeit with the majority of the probability density leaning towards $-1$. This is because a correct final answer typically suggests that the entire solution process is likely accurate, whereas an incorrect final answer may still encompass some correct intermediate steps. Thus, with the backup of MCTS, the $Q$-values of intermediate steps in incorrect solutions may also be updated with a reward of 1 during simulations.

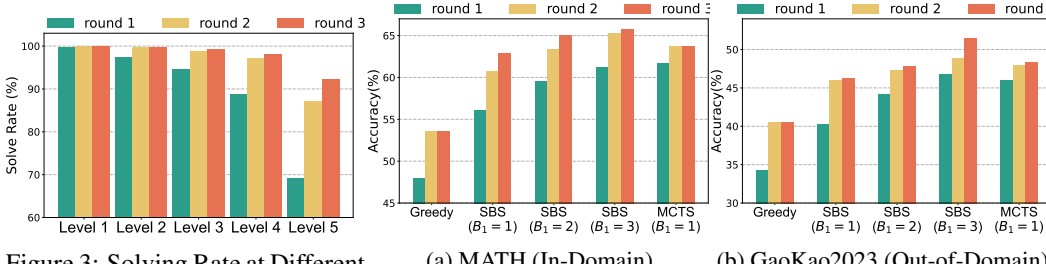

Figure 3: Solving Rate at Different Levels on MATH Training Set

(a) MATH (In-Domain)  (b) GaoKao2023 (Out-of-Domain)

Figure 4: Comparison of Different Inference Strategies.

In the right panel of Figure 5, we plot the $Q$-values distribution on the test set, including both intermediate and terminal steps. The distribution associated with correct solutions exhibits a shape similar to that found in the training set. However, the distribution of incorrect solutions, which are the bad cases of the policy model, demonstrates a bimodal pattern. **(1)** When the value model believes the incorrect solution predicted by the policy model to be incorrect, the $Q$-values cluster around $-1$. **(2)** Conversely, there are instances where the value model erroneously considers an incorrect solution as correct, resulting in another modal towards 1, which represents the bad cases of the value model.

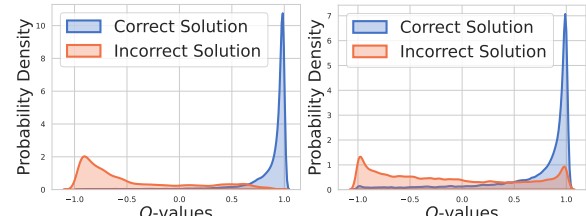

Figure 5: (Left) Fitted distribution of $Q$-values of 3rd round MCTS on the training set. (Right) Fitted distribution of $Q$-values via MCTS inference on the test set.

### 4.6 Analysis 4: Self-evolution on General-purpose and SFT models

We further investigate the potential of two other popular types of LLMs: general-purpose pre-trained models and SFT models. These models represent the scenarios of lacking continual pre-training (CPT) in domain-specific data and supervised fine-tuning (SFT) on high-quality annotated domain data, respectively. We select Llama3 [11] and MARIO [21] as the base models and report the results in Table 4. For a fair comparison, the MARIO is trained on DeepSeekMath-Base-7B rather

Table 4: Additional Results on Llama3 and MARIO. [†]DeepSeekMath-Base-7B. [§]Our designed prompt in 2-shot setting.

| Model | In-Domain | | OOD | |
| --- | --- | --- | --- | --- |
| | GSM8K | MATH | GK2023 | OCW |
| Llama3-base[§] | 40.7 | 18.7 | 12.9 | 2.9 |
| + AlphaMath ($K = 3$) | 59.4 | 36.8 | 27.1 | 6.6 |
| + SBS ($B_1 = 3$) | 71.8 | 41.9 | 31.4 | 10.7 |
| DSM[†] + 27k MARIO data | 78.4 | 56.1 | 41.6 | 25.0 |
| + AlphaMath ($K = 2$) | 80.2 | 58.8 | 48.1 | 31.3 |
| + SBS ($B_1 = 3$) | **88.3** | **68.6** | **54.1** | **42.3** |

than its original Llemma-7B [3]. First, although not proficient in mathematical reasoning, our AlphaMath enhances Llama3's mathematical reasoning capabilities without any annotations, yielding an average improvement of +20 points. Secondly, AlphaMath can significantly enhance the performance of existing SFT models, enabling MARIO to be competitive with and even outperform GPT-4.

### 4.7 Analysis 5: The Effects of Temperature on Step-level Beam Search

We further investigate the effects of temperature during decoding on the performance of inference algorithms. For the greedy strategy, the temperature is consistently maintained at 0, whereas step-level beam search (SBS) and Monte Carlo Tree Search(MCTS) are more significantly influenced by higher temperatures. Therefore, taking step-level beam search ($B_1 = 1$ and $B_1 = 3$) as an example, we obtained the results as illustrated in Figure 6.

**First**, under any temperature setting, the performance of step-level beam search significantly surpasses that of the

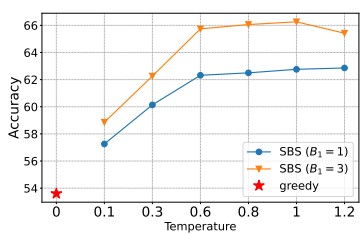

Figure 6: The Effects of Temperature on the performance of SBS.

greedy strategy. This is attributed to the value model effectively assisting the policy model in identifying more effective reasoning paths. **Secondly**, at lower temperatures, the performance of step-level beam search is constrained due to the lack of diversity in the generated solutions. With elevated temperatures, the value model is capable of discerning optimal paths within a more diverse set of solutions, thereby effectively enhancing reasoning performance. **Finally**, with a larger beam width, the model can explore more solutions. Therefore, the performance of $B_1 = 3$ always surpasses that of $B_1 = 1$.

## 5 Related Works

**Solution Annotation in Math.** Recent works [46, 38, 14, 20, 21, 31, 25, 16] on mathematical reasoning have made impressive progress empowered by process-supervised data. However, most existing efforts concentrate on seeking high-quality solutions from domain experts or formidable commercial models, such as GPT-4 [27], which hampers the scalability of methods and escalates the associated expenses. Unlike previous work, only with the help of question-answer pairs, we focus on activating the intrinsic knowledge within LLMs to realize iterative self-evolution and strengthen their utilization of knowledge autonomously, just like humans.

**Value/Reward Model.** Recent studies [7, 8, 22, 44, 49, 42, 41, 12] have demonstrated that process supervision can significantly enhance mathematical reasoning performance. Especially, value model [12, 23, 26] is incorporated into the decoding process, while reward model is the source of the training signal in reinforcement learning [28, 31]. However, these value/reward models require substantial annotated process-supervised data and introduce significant inference latency. In our work, we consider the state values $\widetilde{V}(\mathbf{s}_t)$ from MCTS as supervision signals, which are aligned with the solutions and eliminate the annotation costs. Furthermore, we integrate the value model into the generative model to navigate more effective reasoning paths at minimal cost, thereby providing richer decoding strategies, such as step-level beam search or MCTS.

## 6 Conclusion

In this work, we introduce AlphaMath, a simple iterative training paradigm for leveraging Monte Carlo Tree Search to unleash the potential of a well pre-trained large language model to autonomously enhance its mathematical reasoning capabilities. Furthermore, by applying step-level beam search, the value model can assist the policy model in selecting a more reasonable solution path, rather than solely relying on prior probabilities, which significantly enhances mathematical reasoning capabilities at minimal cost. The experimental results on both in-domain and out-of-domain datasets demonstrate that even without GPT-4 or human-annotated process supervision, AlphaMath remains competitive with or surpasses the performance of the state-of-the-art methods.

## Acknowledgments

This work was supported by Alibaba Research Intern Program.

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

# Contents of Appendix

# A  Dicussion

## A.1  Limitation

Compared to previous works, our AlphaMath achieves comparable or even superior results without annotated high-quality, process-supervised data. However, unlike the game of Go, where the final board configuration directly reflects winning or losing, in mathematical reasoning, we rely on the actual answer as the source of reward. This hinders us from "*AlphaMath really from zero*", an unsupervised algorithm. However, compared to process-supervised data, the acquisition of actual answers is considerably more straightforward. For instance, existing question-answering datasets as well as questions from examinations typically encompass the answers, yet lack annotations for process supervision.

## A.2  Future Work

**Directions for Future Work:** Our research highlights several issues for further exploration:

- **Really from Zero:** In this work, we have demonstrated that a well pre-trained large language model can unleash its potential to identify correct mathematical reasoning processes through the AlpahMath framework, independent of GPT-4 or manually annotated process-supervised datasets. A challenging yet profoundly meaningful future direction is to identify an appropriate reward definition that allows AlphaMath to eliminate dependence on actual answers, thereby achieving really from zero. Notably, this process should avoid introducing additional annotation costs, such as training a reward model to replace the actual answers.

- **A closed-loop self-evolution training framework:** With the question-answer pairs, our AlphaMath framework can realize iterative self-evolution in complex reasoning scenarios, just like humans. In this study, as an initial attempt, we have maintained the same set of question-answer pairs (total only *15k pairs*) in each iteration, which limits the potential of AlphaMath. In the future, we will explore how to automatically obtain such question-answer pairs from the Internet, which could facilitate the development of a closed-loop self-evolution framework for AlphaMath. In this setup, the LLM automatically acquires question-answer pairs from the Internet and then autonomously enhances its reasoning capabilities through our AlphaMath framework, thereby achieving complete independence from human intervention.

- **Explore beyond mathematical reasoning:** Since mathematical reasoning tasks involve complex, symbolic multi-step reasoning, we primarily choose them as an example to investigate the effectiveness of AlphaMath. However, our proposed AlphaMath has the potential to be broadly applied to any task that can be evaluated against actual answers. In future work, we plan to expand its application to a broader range of tasks.

# B  Supplementary Experiments and Analysis

## B.1  More Results of Inference Strategies

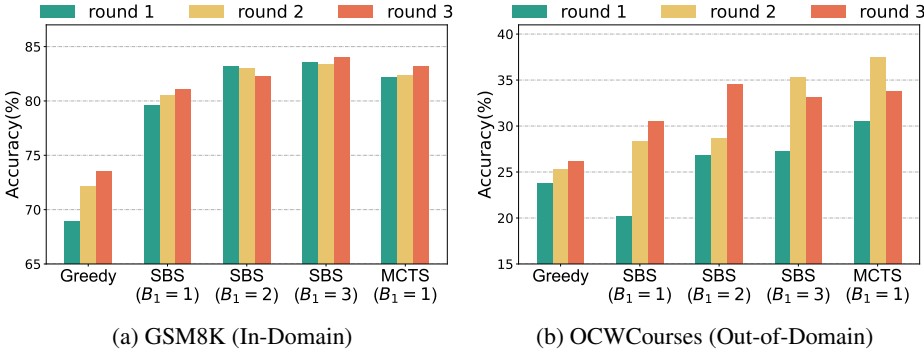

(a) GSM8K (In-Domain)        (b) OCWCourses (Out-of-Domain)

Figure 7: Comparison of Different Inference Strategies (Other datasets).

In this experiment, we can draw similar conclusions as in Figure 4. With the progress of iteration, there is a significant enhancement in the model's performance, especially between the first and second rounds, as shown in Figure 7. Furthermore, we observe that the performance of various inference strategies on the OCWCourses slightly differs from the other three datasets. This variation can be attributed to the fact that OCWCourses is a mathematical dataset in the fields of physics and chemistry. Nonetheless, our method still significantly enhances the model's reasoning capabilities on such datasets overall.

## B.2 More Analysis of Value Model

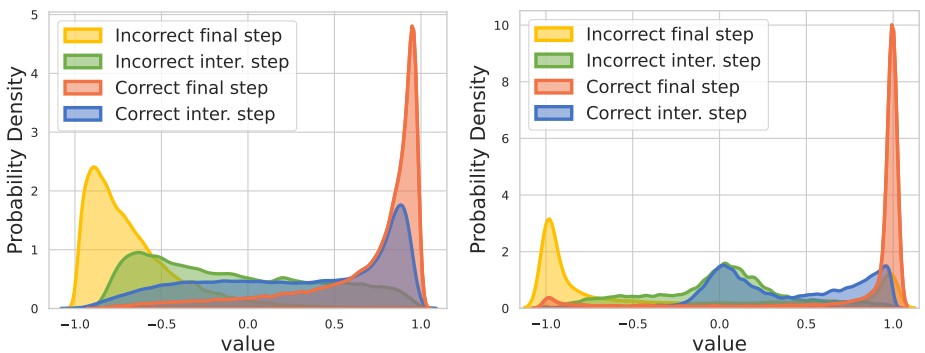

Figure 8: (Left) Fitted distribution of **value predictions** of sampled solutions on the training set. (Right) Fitted distribution of **value predictions** of sampled solutions on the test set. "Incorrect inter. step" denotes the state value $V(\mathbf{s})$ of an intermediate step within an incorrect solution.

In addition to the discussion regarding the value model in Section 4.5, we have also specifically analyzed the overall distribution of the predicted state values $V(\mathbf{s})$ by the value model for both the intermediate and final steps in correct/incorrect solutions, as illustrated in Figure 8. "Final step" refers to the scoring of the entire solution in the last step, representing the value model's overall assessment.

In the left panel of Figure 8, we plot the fitted distribution of the state values for both intermediate and final steps as predicted by the value model, during the process of solution generation on the training set. For correct and incorrect solutions, the value model's overall assessment is highly accurate, which is distinctly skewed towards 1 and -1, respectively. Notably, "Incorrect inter. Step" represents the intermediate steps of an incorrect solution, rather than incorrect intermediate steps. Therefore, "Incorrect inter. Step" may also contain some correct processes, which explains why its distribution crosses over 0. Overall, the distribution of the value model on the training set aligns very well with intuition, which aids in identifying higher-quality solutions in MCTS.

In the right panel of Figure 8, we plot the distribution of state value $V(\mathbf{s})$ predicted by the value model on the test set. It can be clearly seen that the value model accurately distinguished between correct and incorrect solutions, which explains why the performance of step-level beam search significantly surpasses that of greedy inference. The value model aids the policy model in navigating more efficient solutions, rather than relying solely on prior probabilities. Additionally, due to the fact that incorrect solutions may contain correct steps, their distribution is primarily concentrated near 0. The intermediate steps of correct solutions exhibit a bimodal distribution, with peaks concentrated near 0 and 1. This can be attributed to the fact that even correct solution steps may contain some errors, such as coding mistakes. Therefore, in conjunction with the analysis in Section 4.5, we believe that our value model can effectively distinguish between correct and incorrect solutions, aiding the policy model in finding better solution paths.

## B.3 More Analysis of Problem Solving Rate of MCTS in Each Round

In this experiment, we evaluate the successful solving rate of MCTS across various rounds. Utilizing the MATH dataset, which categorizes each problem by difficulty level and subject type, we compute the problem-solving rate across different categories and difficulty levels. For our training set, we

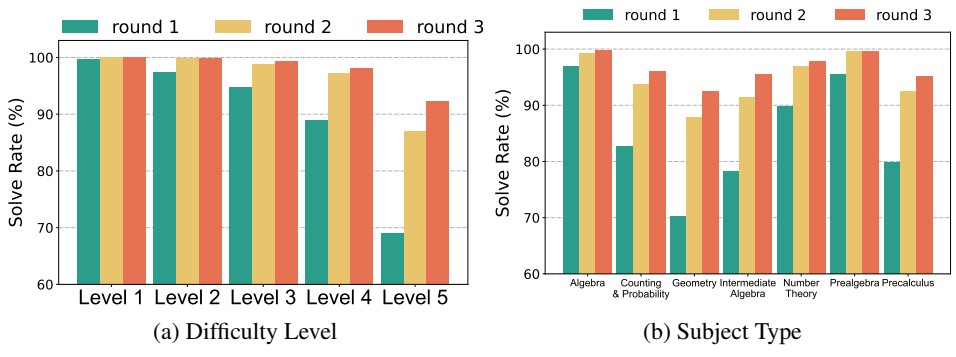

(a) Difficulty Level

(b) Subject Type

Figure 9: Problem Solving Rate on MATH Training Set

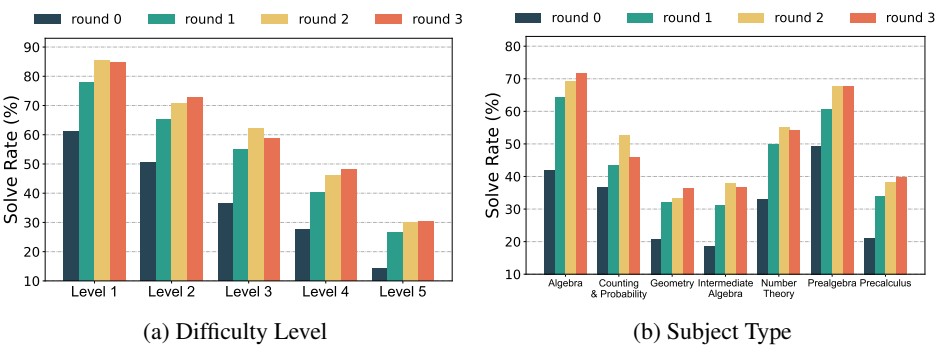

(a) Difficulty Level

(b) Subject Type

Figure 10: Problem Solving Rate on MATH Test Set

count the instances wherein problems are successfully solved along any of the paths within the 10 constructed trees. As illustrated in Figure 9a, it becomes evident that MCTS achieves greater success in solving more challenging problems in subsequent rounds. Similarly, Figure 9b indicates that, in later rounds, MCTS consistently demonstrates an improved capability to solve a broader array of problems across different subjects.

For the test set depicted in Figure 10, we include the results from round 0, which correspond to the performance of our "prompt 2-shot" in Table 2. Unlike the training set, we observe that the improvement observed in round 3 is not consistent across different levels and subjects, even though the overall accuracy is slightly increased. In fact, for easier problems, the performance in round 3 actually declines. This is the reason we terminate our iterative training process after round 3.

## B.4  Problem Solving Rate for Each LLM in Training Set

Table 5: Solving Rate for Each LLM in Training Set. §Since MARIO is a SFT model and already possesses the capability to follow instructions, we opted to skip its first round.

| Model | Round 1 | | Round 2 | | Round 3 | |
|---|---|---|---|---|---|---|
| | GSM8K | MATH | GSM8K | MATH | GSM8K | MATH |
| DeepseekMath-base-7B [31] | 97.24% | 83.93% | 99.90% | 93.73% | 99.94% | 95.61% |
| Llama3-base-8B [11] | 94.48% | 78.42% | 99.07% | 89.77% | 99.92% | 94.50% |
| MARIO§ [21] | - | - | 99.91% | 94.51% | 99.97% | 94.79% |

We further discuss the problem solving rates of different models in each round, as shown in Table 5. **First**, given that the problems in the GSM8K dataset are relatively simple, the corresponding solution rates are higher, even in the first round. Despite the challenging nature of MATH, its problem-solving rate increases with more iterations, indicating continuous improvement in the model's performance. **Secondly**, there are also noticeable differences in the problem-solving rates between different models.

Since the SFT model (MARIO) is fine-tuned on high-quality data, it exhibits the best performance. Furthermore, the math domain-specific pre-trained LLM (DeepseekMath-base-7B) significantly outperforms general-purpose pre-trained LLM (Llama3). This phenomenon is intuitive because domain-specific pre-trained LLMs possess more specialized knowledge. **Finally**, we note that in the third round, the solution rates of each model are quite similar. However, the final performance of the models differs significantly, as shown in Table 4. This discrepancy can be attributed to variations in the quality of solutions generated by different models. Generally speaking, the more relevant knowledge is embedded within LLMs, the higher the quality of problem-solving solutions that will be generated. This explains why domain-specific pretrained models significantly outperform general-purpose pretrained models. However, the advantage of our proposed AlphaMath lies in its ability to significantly enhance the performance of existing models without relying on high-quality data annotated by GPT-4 or humans. Even with a weaker general-purpose pre-trained model, AlphaMath achieves a remarkable +20 points improvement, as shown in Table 4. Furthermore, our AlphaMath enables a domain-specific pre-trained model to achieve comparable or even superior results compared to state-of-the-art SFT models.

## C   Implementation Details

### C.1   Definitions of various elements in MCTS

**State**   The state $\mathbf{s}_t$ is defined as a partial solution, consisting of the initial prompt (question) and all actions taken along the Monte Carlo tree search path from the root node to the current node, as shown in Eq. 1.

**Node**   Nodes are used to record information, such as the action (step) $\mathbf{a}_t$, the value predicted by the value model $V_\phi$, the state-action value $Q$ from MCTS, depth, visiting counts, and etc. Each node is defined to only contain a single step.

**Action (Steps)**   Following Liao et al. [21], we define two types of actions (steps) $\mathbf{a}_t$: $\mathcal{C}$-steps and $\mathcal{A}$-steps. Each node contains only one type of action, and $\mathcal{A}$-steps typically appear at the end, as shown in Figure 14. $\mathcal{C}$-step represents code execution, which is composed of textual analysis, code snippets, and execution results. The textual analysis and code snippets are generated by the policy model (LLM), while the execution results are the outputs returned by the Python code interpreter. $\mathcal{A}$-step represents the summary of the answer, which is composed of text analysis and predicted answers. Both the text analysis and predicted answers are generated by the policy model. We organize these two steps in the following XML format:

> **$\mathcal{C}$-step**
>
> ```
> <step>\n<p>\n{textual analysis}\n</p>\n\n{code
> snippets}\n\n<p>\n{code output}\n</p>\n</step>
> ```

> **$\mathcal{A}$-step**
>
> ```
> <step>\n<p>\n{textual analysis}\n</p>\n<p>\nFinal
> Answer:{predicted answer}\n</p>\n</step>
> ```

### C.2   Solution Filtering Algorithm

After solution generation via MCTS, we randomly sample the correct and incorrect solutions of each question for training. During this process, we found that the generated solutions might suffer from issues such as hallucinations [17]. Hence, we propose a solution filtering algorithm to optimize the solution selection.

Algorithm 3 outlines the process of our solution filtering algorithm. We initially deduplicate and remove the solutions where code errors are present across all steps (Lines 3-5). As indicated in Figure 11, the solutions that present code errors in all steps yet yield a correct final answer are

**Algorithm 3** Solution Filtering

---

**Require:** Sampled solutions $\mathcal{S}$.
**Ensure:** Candidate correct solutions $\mathcal{S}_c$, candidate incorrect solutions $\mathcal{S}_e$.

1:  $\mathcal{S}_c = [\,], \mathcal{S}_e = [\,]$                                                   ▷ *Initialization*
2:  **for** $s$ in $\mathcal{S}$ **do**
3:     **if** $s$ not in $\mathcal{S}_c$ or $s$ not in $\mathcal{S}_e$ **then**               ▷ *De-duplication*
4:         **if** Code Errors persist across All Steps in $s$ **then**
5:             **continue**        ▷ *Eliminating solutions where errors permeate all steps.*
6:         **if** $s$ is incorrect solution **then**
7:             Add $s$ to $\mathcal{S}_e$                   ▷ *InCorrect Solution*
8:         **else**
9:             flag $\leftarrow$ False
10:            **for** each output of code $o$ in $s$ **do**
11:                **if is\_equiv**($o$, predict answer in $s$) **then** ▷ *is\_equiv is the evaluation toolkit [47].*
12:                   flag $\leftarrow$ True
13:                   **Break**
14:            **if** flag **then**
15:                Add $s$ to $\mathcal{S}_c$             ▷ *Level 1 Correct Solution*
16:            **else**
17:                **if** code is correct at every step in $s$ **then**
18:                   Add $s$ to $\mathcal{S}_c$         ▷ *Level 2 Correct Solution*
19:                **else**
20:                   Add $s$ to $\mathcal{S}_c$         ▷ *Level 3 Correct Solution*

---

evidently hallucinations. To preserve the diversity of incorrect solutions, we refrain from excessive processing, which aids the value model in being exposed to a wide variety of incorrect solutions (Lines 6-7). Furthermore, to filter out higher-quality yet diverse correct solutions, we categorize them into different levels. We consider the solutions where the predicted answer is consistent with the code output as level 1 correct solutions because the model has generated code and understood the output correctly (Lines 9-15). Then, we observe that for some questions, the quality of a solution cannot be solely gauged by the consistency between the code's output and the predicted answer (for which we provide an example in Figure 13). Therefore, we view the solutions with entirely correct code as level 2 correct solutions (Lines 17-18). Generating correct code to some extent proves that the model has a correct understanding of this question. Finally, we consider the remaining correct solutions as level 3 solutions (Line 20).

In general, Algorithm 3 offers a method for swiftly filtering solutions. By categorizing solutions into different levels, we prioritize higher-level solutions for each problem. We provide several examples to intuitively illustrate the necessity and rationality of Algorithm 3, as shown in Figure 11 12 13.

---

(a) An example of code errors persist across all steps:

```
<question>
What is the sum of all positive integers $\nu$ for which $\mathop{lcm}[\nu,20]
=60$?
</question>
<step>
<p>
To find the sum of all positive integers $\nu$ for which $\mathop{lcm}[\nu,20]
=60$, we need to find the values of $\nu$ that satisfy the equation. We can
start by simplifying the equation by dividing both sides by 20, which gives us
$\mathop{lcm}[\nu,1]=3$. Since $\mathop{lcm}[1,1]=1$, we can see that $\nu$ must
be equal to 3. Therefore, the sum of all positive integers $\nu$ for which
$\mathop{lcm}[\nu,20]=60$ is 3.
</p>

```python
from sympy import lcm
```

```
# Define the equation
equation = lcm(nu, 20) == 60
# Solve the equation
solution = solve(equation, nu)
# Find the sum of the solutions
sum_of_solutions = sum(solution)
print(sum_of_solutions)
```

<output>
AttributeError: 'int' object has no attribute 'is_commutative'
</output>
</step>
<step>
<p>
From the result, we can see that there is an error. This is because the `solve`
function is not able to solve the equation with the `lcm` function. We need
to find a different way to solve the equation.
</p>

```python
from sympy import factorint, lambdify
# Define the equation
equation = lcm(nu, 20) == 60
# Factorize the numbers in the equation
factorized_equation = equation.subs(lcm(nu, 20), lcm(factorint(nu),
        factorint(20)))
# Simplify the equation
simplified_equation = simplify(factorized_equation)
# Evaluate the equation
evaluated_equation = eval(simplified_equation.str)
# Solve the equation
solution = solve(evaluated_equation, nu)
# Find the sum of the solutions
sum_of_solutions = sum(solution)
print(sum_of_solutions)
```

<output>
AttributeError: 'bool' object has no attribute 'subs'
</output>
</step>
similar error repeated in subsequent steps...
<step>
<p>
From the result, we can see that there is an error. This is because the `subs`
function is not able to substitute the `lcm` function with the `factorint`
function. We need to find a different way to factorize the numbers in
the equation.
</p>
<p>
Final Answer: $126$
</p>
</step>
```

Figure 11: An example of code errors persists across all steps. In this example, while the model was adjusting the code to rectify coding errors, each step of the solution generated an error. Nevertheless, in the final step, the model still produced the correct answer. This is a classic **hallucination**. Hence, we discard such solutions in Algorithm 3.

### C.3 Parameter Details

**For the solution generation via MCTS**, we set $c_{\text{puct}} = 1.25$, set the temperature between $0.6$ to $1.2$, limit the maximum tree depth to 8, and allow each node to expand 5 child nodes ($B_2 = 5$),

(b) An example of level 1 Correct Solution

```
<question>
What is the value of $525^2 - 475^2$?
</question>
<step>
<p>
To find the value of $525^2 - 475^2$, we can use the difference of squares
formula: $a^2 - b^2 = (a + b)(a - b)$.
</p>

```python
a = 525
b = 475
# Use the difference of squares formula
# value = (a + b) * (a - b)
print(value)
```

<output>
50000
</output>
</step>
<step>
<p>
The result from the Python code gives us the value of $525^2 - 475^2$ as $50000$.
</p>
<p>
Final Answer: $50000$
</p>
</step>
```

Figure 12: An example of the level 1 correct solution. In this example, the model generates the correct code and provides a correct answer based on the output of this code. We regard such solutions as dependable and categorize this type of solution as the level 1 correct solutions.

and a maximum of 40 simulations ($N = 40$). Since MARIO [21] already has the ability to follow instructions, we set $K = 2$ for MARIO. For DeepseekMath-Base [31] and Llama3 [11], we set $K = 3$. In each round, we build 10 trees for each question-answer pair and randomly sample at most 4 correct and 4 incorrect solution processes for training. In this setting, the ratio of positive to negative examples is approximately 1:1, and the count of positive examples varies between 57k to 59k for each round.

**For supervised fine-tuning**, we set the learning rate of 4e-5, batch size of 1024, the weight of the value loss to 0.01 or 0.0005 (for Llama3 [11]), and train the model for 10 epochs. We employ the AdamW optimizer [24] and the cosine learning rate scheduler with the warmup rate set to 0.03. More hyperparameter details can be found in Table 6.

**For baselines**, the results recorded in Table 2 come from corresponding published papers.

### C.4 Policy-Value model Details

Unlike previous work [49, 44, 12, 5], the value model is trained separately to assist the policy model. In this study, we integrate the value model into the policy model by appending a linear layer, as illustrated in Figure 1. Since most of the existing LLMs adhere to decode-only architecture, we utilize the last token as the representation of the entire reasoning step, similar to the role of "[CLS]" token in BERT [10]. In our case, it is typically "</step>", which ensures that the representation of the reasoning step will not be affected by the last token itself. The value model and the policy model share the majority of parameters, that is, they share the understanding of the reasoning steps. The value model assesses the expected returns based on the current reasoning step, while the policy model generates the next token.

```
<question>
How many $y$-intercepts does the graph of the parabola $x = 2y^2 - 3y + 7$ have?
</question>
<step>
<p>
To find the $y$-intercepts of a parabola, we need to set $x = 0$ and solve for
$y$. In this case, we have $x = 2y^2 - 3y + 7$, so we need to solve the quadratic
equation $2y^2 - 3y + 7 = 0$.
</p>

```python
from sympy import solve, symbols
# define variables
y = symbols('y')
# solve the quadratic equation
solution = solve(2 * y**2 - 3 * y + 7, y)
print(solution)
```


<output>
[3/4 - sqrt(47)*I/4, 3/4 + sqrt(47)*I/4]
</output>
</step>
<step>
<p>
The quadratic equation $2y^2 - 3y + 7 = 0$ has no real solutions, which means
the parabola has no $y$-intercepts. We can also see this from the complex
solutions for $y$ in the previous step.
</p>
<p>
Final Answer: $0$
</p>
</step>
```

Figure 13: An example of the level 2 correct solution. In this example, the consistency between the code's output and the answer does not adequately measure the quality of the solution. Therefore, we categorize solutions that are entirely correct in terms of the code as the level 2 correct solutions.

Table 6: Key hyperparameters of AlphaMath

| Hyperparameter | Value |
| --- | --- |
| $c_{puct}$ | 1.25 |
| $K$ | 3 or 2 (for MARIO [21]) |
| Weight of value loss $\beta$ | 0.1 or 0.0005 (for Llama3 [11]) |
| $B_1$ | $\{1, 3\}$ |
| $B_2$ | 5 |
| Simulations $N$ | 40 |
| Temperature | $\{0.6, 1.0, 1.2\}$ |
| max depth (max steps) $T$ | 8 |
| Batch size | 1024 |
| Optimizer type | AdamW [24] |
| Learning rate | 4e-5 |
| lr scheduler type | cosine |
| Warmup ratio | 0.03 |
| Epochs | 10 |
| Weight decay | 0. |

## C.5 Datasets Details



Table 7: Datasets Statistics

| Dataset | OOD? | # Training | # Test |
|---|---|---|---|
| GSM8K [7] | In-Domain | 7473 | 1319 |
| MATH [15] | In-Domain | 7500 | 5000 |
| GaoKao2023 [21] | OOD | - | 385 |
| OCWCourses [19] | OOD | - | 272 |



Table 7 describes the statistics of datasets in detail. The division of the training and test set follows the previous work [7, 15]. GSM8K [7] is a multi-step mathematical reasoning dataset comprising high-quality, diverse grade school math word problems, created by human problem writers. MATH [15] is a dataset of challenging competitive mathematics problems. GaoKao2023 [21] is a collection of mathematics problems from the 2023 Chinese National College Entrance Examination, the 2023 American Mathematics Competitions, and the 2023 American College Testing, while OCWCourses [19] comprises a collection of 272 STEM problems aimed at the undergraduate level, requiring multi-step reasoning for most questions.

## C.6 Experiment Environments

All experiments were conducted on Ubuntu 22.04 equipped with 8 * NVIDIA A100 GPUs. Our code mainly depends on Python 3.11[4] and PyTorch 2.1.2[5]. We use our customized *Llama Factory* [48] as the training framework and our customized *vLLM* [18] as the inference framework[6]. We trained all models with *DeepSpeed ZeRO Stage2* [29] and *Flash-Attention 2* [9]. The pre-trained language models are derived from *HuggingFace*[7].

# D Case Study

**Solution Generation in Round 1**   Figure 14 illustrates an example of solution generation on the MATH dataset via MCTS in round 1. We guide the pretrained model, such as DeepseekMath-Base [31], to generate solutions in the form of Thought/Action/Action Input/Observation, as shown in Sec. F.1. For clarity of presentation, we only illustrate a subset of the nodes of the original Monte Carlo tree in Figure 14. As shown in Figure 14, the path $(a) \rightarrow (c) \rightarrow (f)$ represents a correct solution, whereas the other solutions contain errors to some degree. Node (b) attempts to solve the problem in a single step and proposes a correct thought. However, minor errors in the coding process ("k = 7 **2 * 3**3" was mistakenly written as "k = 7 * 3**3") led to mistakes in all subsequent steps. Node (d) attempts a different approach from node (c), specifically trying to solve for $a$ first, then proceeding to solve for $a^2$. Although this process is more redundant compared to that of node (c), it is nonetheless a correct approach. However, in subsequent steps, we encountered errors of various forms. Firstly, within the node (g), the model mistakenly treats the output for (d) as equivalent to $a^2$, leading directly to an output. At node (h), the model opts to calculate a relying on its capabilities; however, this results in a numerical error. From a holistic perspective, we observe that, aided by MCTS, the pretrained model attempts to solve the problem through various approaches. During this process, we naturally excavate the knowledge embedded within the model, thereby reinforcing the model's understanding and application of this knowledge in subsequent training iterations. Furthermore, we collect the $Q$-values along the path to aid the model in enhancing its judgment of the correctness of the reasoning process.

**Solution Generation in Round 3**   Figure 15 illustrates an example of solution generation via MCTS in round 3. Compared to round 1, the quality of the model in round 3 has significantly improved, resulting in more correct solutions. We observe that the code quality in round 3 is superior, containing more comprehensive annotations. This signifies an enhancement in the model's understanding of the question and its coding proficiency. Furthermore, as illustrated by node (b), the model attempts to resolve the question in a single step, yielding solutions that are both shorter and more efficient.

---

[4]https://www.python.org/

[5]https://pytorch.org/

[6]We have released our customized framework in our Github Repository.

[7]https://huggingface.co/

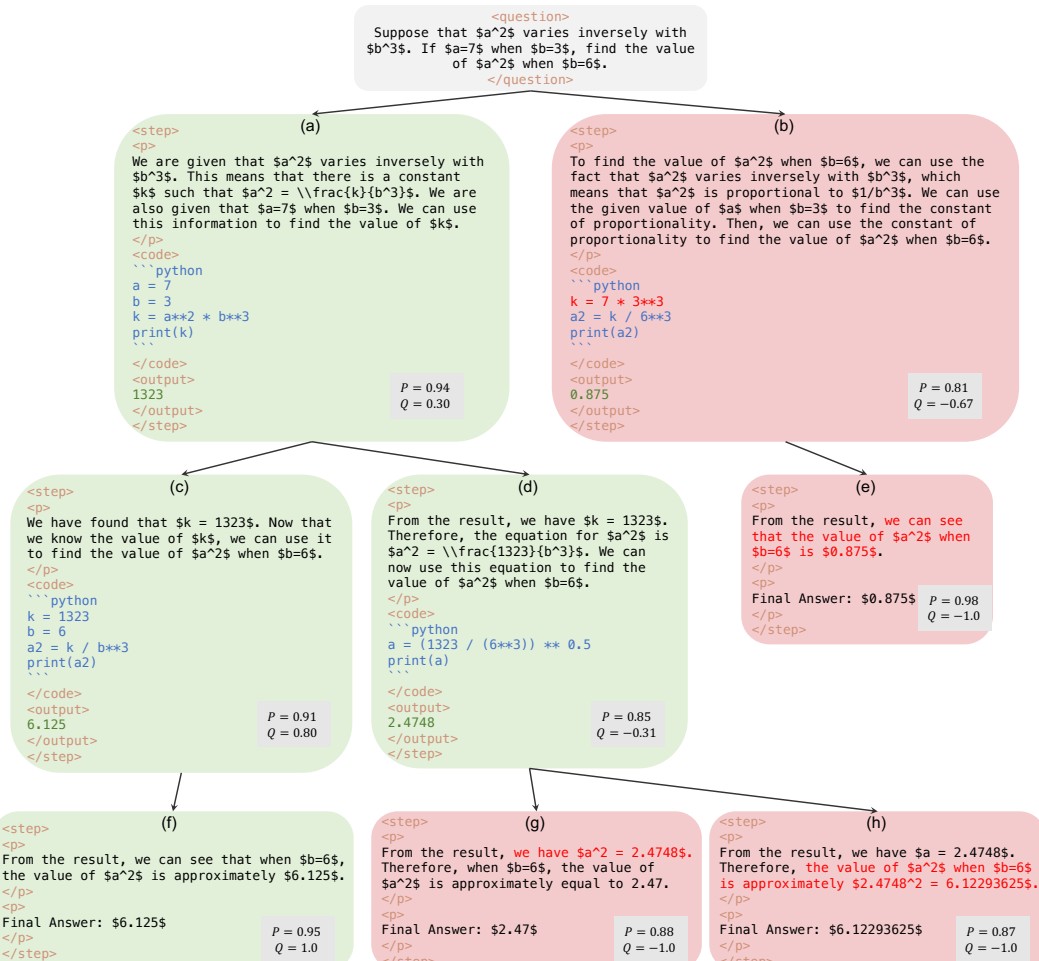

Figure 14: Example of Solution Generation via MCTS in Round 1. The green and red areas represent correct and incorrect nodes, respectively. The red text segment indicates a specific error in the solution. In round 1, due to the value head being randomly initialized, the estimated values are not accurate; therefore, we have not presented the estimated value here. For the sake of clarity in our demonstration, we only display a subset of the original Monte Carlo tree and present each node in XML format (as detailed in Appendix F.2), even though the format utilized in round 1 was Thought/Action/Action Input/Observation (as detailed in Appendix F.1).

## E Error Analysis

In this section, we conducted a comprehensive error analysis on the MATH dataset to foster future work. We randomly select 100 error solutions for analysis. We find the following three types of errors.

**(1) Numerical Errors (53%).**  This is one of the main sources of errors. Within this category of errors, the model usually possesses the correct thought to solve the problem, but an error occurs during the numerical computation. The most common types of numerical computation errors arise from incorrect code generated by the model, which we will not discuss in detail. We have identified errors caused by numerical precision issues in Python, which we refer to as "numerical precision errors". Here is an example:

```
Question: The first two terms of a sequence are $a_1 = 1$ and $a_2 =
\frac{1}{\sqrt3}$. For $n\ge 1$, \begin{align*} a_{n + 2} = \frac {a_n +
a_{n + 1}}{1 - a_na_{n + 1}}. \end{align*}What is $a_{2009}$?
```

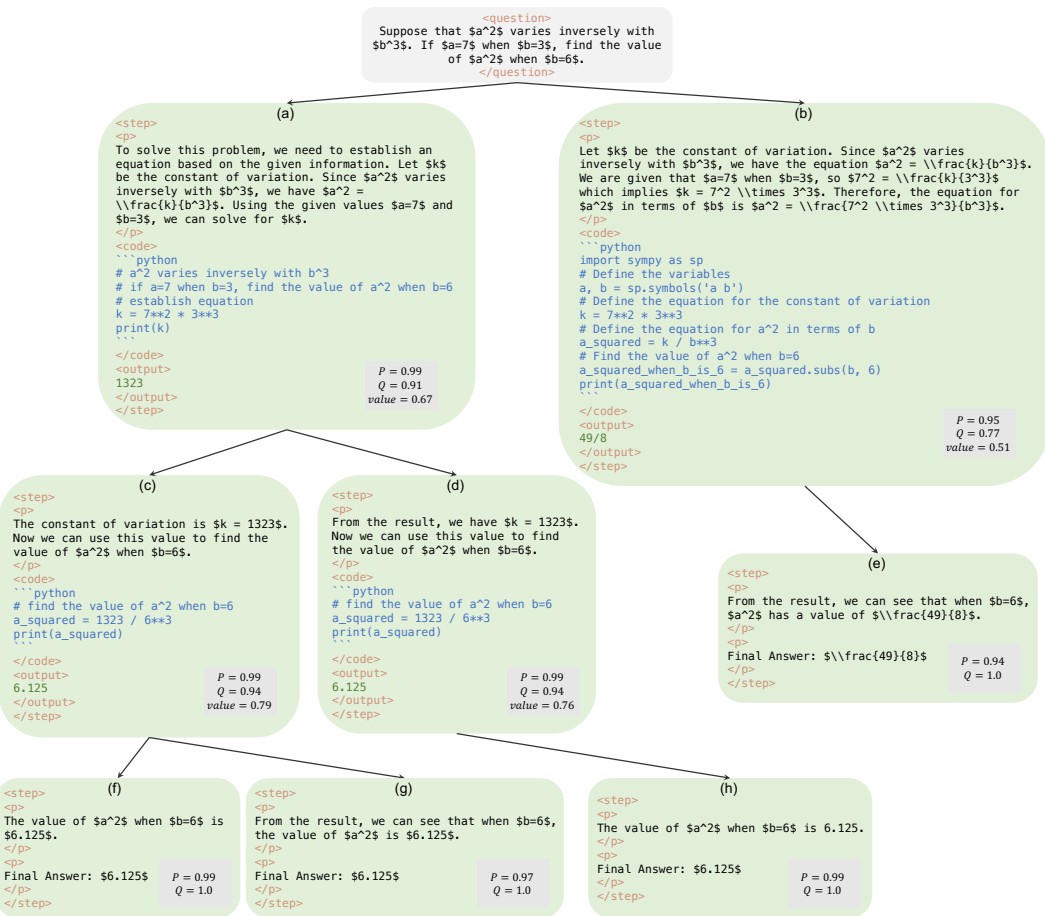

Figure 15: Example of Solution Generation via MCTS in Round 3.

The actual answer: 0

The thought of LLM: To find $a_{2009}$, we need to calculate the first 2009 terms of the sequence. However, the sequence is defined recursively, so we need to find a way to calculate the next term based on the previous two.

The code written by LLM:
```python
import math
a1 = 1
a2 = 1 / math.sqrt(3)
for i in range(2, 2009):
    an = a2
    an_1 = a1
    a2 = (an + an_1) / (1 - an * an_1)
    a1 = an
print(a2)
```

The output of above code: -2.7267039150707353

In this example, the model possesses the correct thought of problem-solving and generates the correct Python code. But an incorrect answer was still produced. To ensure numerical precision, we found that the correct implementation should simplify the equation using "sympy", as shown below:

```python
import sympy as sp
a1 = 1
a2 = 1 / sp.sqrt(3)
```

```
for i in range(2, 2009):
    an = a2
    an_1 = a1
    a2 = sp.simplify((an + an_1) /(1-an*an_1))
    a1 = an
print(a2)
```

The output of above code:  0

**(2) Logical Errors (45%).**   This is another major source of errors. In this type of error, the model usually ignores some details in the question, resulting in incorrect problem-solving approaches. Indeed, humans can also make similar mistakes due to carelessness. Here is an example:

Question: If $x$ and $y$ are numbers such that $(x + 3)(y-5) = 0$, what is the smallest possible value of $x^2+ y^2$?

The thought of LLM:  To calculate the value of $x^2+ y^2$, we need to solve the equation $(x + 3)(y-5) = 0$. After finding the solutions for $x$ and $y$, we need to calculate $x^2+ y^2$ by substituting these values into the expression $x^2 + y^2$.

The code written by LLM:
```
from sympy import symbols, solve
x, y = symbols('x y')
eq1 = Eq(x + 3, 0)
eq2 = Eq(y - 5, 0)
solution1 = solve(eq1, x)
solution2 = solve(eq2, y)
x_value = solution1[0]
y_value = solution2[0]
result = x_value**2 + y_value**2
print(result)
```

In this example, the model overlooked the requirement in the question to find the smallest possible value, resulting in the incorrect code and final answer.

**(3) Other Errors (2%).**   Aside from the two common types of errors mentioned above, we have also discovered other errors, which may stem from the ambiguity in the question statements or inaccuracies in the actual answer. Here is an example:

Question: For which values of $k$ does the quadratic $kx^2 - 3kx + 4k + 7 = 0$ have real roots?

The actual answer:  [-4,0)

The thought 1 of LLM:  To find the values of $k$ for which the quadratic equation $kx^2 - 3kx + 4k + 7 = 0$ has real roots, we need to consider the discriminant of the equation, which is $b^2 - 4ac$. If the discriminant is positive, the equation has two distinct real roots, and if it is zero, the equation has one real root.

The code 1 written by LLM:
```
from sympy import symbols, Eq, solve
k = symbols('k')
discriminant = (-3*k)**2 - 4*k*(4*k + 7)
print(discriminant)
```

The output of above code:  9*k**2 - 4*k*(4*k + 7)

The thought 2 of LLM:  The discriminant is $9k^2 - 4k(4k + 7)$. To find the values of $k$ for which the discriminant is non-negative, we need to solve the inequality $9k^2 - 4k(4k + 7) \geq 0$.

The code 2 written by LLM:
```python
from sympy import symbols, Eq, solve
k = symbols('k')
inequality = 9*k**2 - 4*k*(4*k + 7) >= 0
solutions = solve(inequality, k)
print(solutions)
```

The output of above code:  (-4 <= k) & (k <= 0)

The predicted answer by LLM:  $[-4, 0]$

In this example, the model correctly solved the problem. However, due to the inaccuracy of the actual answer, the evaluation tool mistakenly deemed the model's predicted answer as incorrect.

## F   Prompts

We provide instructions and examples for generating solutions and performing reasoning. For the solution generation through MCTS in round 1, the pre-trained models such as DeepseekMath-base [31] may not adhere to instructions. Therefore, we employ few-shot learning to guide the model towards producing outputs in the format of Thought/Action/Action Input/Observation. Subsequently, we train the model in our SFT XML format and proceed with both the solution generation and reasoning in this format. We provide specific details regarding two formats of prompts as follows:

### F.1   Prompt Example of MCTS in Round 1

We employ few-shot learning to steer pre-trained models toward performing mathematical reasoning, ensuring the output conforms to the format of Thought/Action/Action Input/Observation. As the following example shows, the text in black is prompt, and the text in red is model generation.

```
You are a powerful agent with broad math knowledge and great Python
programming skills, but not good at accurate calculation on math equations.
Answer the math questions as best you can. You have access to the following
tool:

Python_interpreter: A Python shell. Use this to execute python commands.

!!!Remember:

1. When solving math problem, you should think step by step, where each step
includes 4 mini-steps Thought/Action/Action Input/Observation.

2. If some step requires accurate calculation (including but not limited to
numerical calculation and symbolic deduction), you should write Python code and
execute for accurate result.

3. The most related Python packages include `math`, `sympy`, `scipy` and `numpy`.

4. Please use the following template.

Question: the input question

Thought: the text analysis, and list the math equations if necessary

Action: the action to take, should be `Python_interpreter`, or None

Action Input: the Python Code in markdown format (if Action is None, it is None),
e.g.,
```python
import math
theta = math.pi / 4
some_var = math.cos(theta)
```
```

```
print(some_var)
```

Observation: the result of the action

... (this Thought/Action/Action Input/Observation can repeat N times)

Thought: the final analysis of the answer

Final Answer: the concise answer without verbose context

Here are some examples:

<examples here>

Now! It's your turn.

Question: Ben works 8-hour shifts in a furniture shop. It takes him 5 hours to build 1 rocking chair.  How many chairs can he build in 10 days?

Thought: To calculate the number of chairs Ben can build in 10 days, we need to first determine how many hours he would work in 10 days and then use the given information to calculate the number of chairs he can build in those hours.

Action: python_interpreter

Action Input: ```python
hours_per_day = 8
total_days = 10
hours_in_10_days = hours_per_day * total_days

chairs_per_hour = 1 / 5
chairs_in_10_days = hours_in_10_days * chairs_per_hour
print(chairs_in_10_days)
```

Observation: 16.0

Thought: I have calculated the number of chairs Ben can build in 10 days. The number of chairs is 16.

Final Answer: 16

## F.2 Prompt Example of MCTS after Round 1

To train the SFT model in executing mathematical reasoning, we utilize an XML format alongside zero-shot learning. This approach is adopted because the math-related pre-training corpora are predominantly harvested from the Internet, where HTML tags serve to distinguish various types of content, including text, equations, and code snippets. As the following example shows, the text in black is prompt, and the text in red is model generation.

```
<question>Haley grows at the rate of 3 inches every year. If she is currently
20 inches tall, what will be her height after 10 years?</question>
<step>
<p>
To calculate Haley's height after 10 years, I need to add 10 times the growth
rate of 3 inches to her current height.
</p>

```python
current_height = 20
```

```
growth_rate = 3
years = 10
future_height = current_height + (growth_rate * years)
print(future_height)
```

<output>
50
</output>
</step>
<step>
<p>
I have calculated Haley's height after 10 years. Haley will be 50 inches tall
after 10 years.
</p>
<p>
Final Answer: $50$
</p>
</step>

