# OpenReview forum: "AlphaMath Almost Zero: Process Supervision without Process"
_NeurIPS.cc/2024/Conference — NeurIPS 2024 poster_

### Official Review · Reviewer_5iMV · 2024-07-10

**Soundness:** 2
**Presentation:** 3
**Contribution:** 2
**Rating:** 4
**Confidence:** 4

**Summary:**

The paper proposes an innovative framework named AlphaMath, aimed at enhancing the mathematical reasoning capabilities of large language models (LLMs) without relying on expensive human annotations from domain experts or GPT-4. The authors leverage Monte Carlo Tree Search (MCTS) to allow a pre-trained LLM to autonomously generate process supervision and step-level evaluation signals, integrating a value model with the LLM. At the inference time, the authors propose a step-level beam search strategy, which assists the policy model in navigating more effective reasoning paths. Experimental results demonstrate that AlphaMath achieves comparable or superior results to previous state-of-the-art methods on various datasets without process supervision.

The application of MCTS in LLMs is now somewhat popular, with several highly relevant papers, such as [1] https://arxiv.org/pdf/2406.06592, [2] https://arxiv.org/pdf/2405.00451 and [3] https://arxiv.org/pdf/2309.17179] (note that [1, 2] are published after neurips deadline, so no need to perform comparison, It's just added for reference.)

**Strengths:**

1.The proposed method effectively eliminates the necessity for human or GPT-4 annotation, resulting in significant efficiencies in both financial and temporal aspects.
2. The utilization of Monte Carlo Tree Search (MCTS) can enhance the performance of LLMs in reasoning tasks by facilitating the search for optimal paths. Additionally, training with trajectories derived from MCTS is a commendable approach.
3. The introduction of step-level beam search presents a promising strategy to reduce computational costs.

**Weaknesses:**

1. The paper's results are all program-aided, a fact not clearly identified in the text. Consequently, the comparison with non-program-aided LLMs may not be fair enough.
2. The benchmark only includes GPT models in the 'Proprietary Models' section. It would be beneficial to include other LLMs such as Gemini and Claude for a more robust evaluation.
3. An image illustrating the Step-Level Beam Search (SBS) would greatly aid in explaining the algorithm.

**Questions:**

Check other sections.

**Limitations:**

1. The framework does not generate a reward model that can facilitate other LLMs, which potentially limits its applicability.
2. The framework cannot be extended to tasks with open-ended outputs, limiting its potential use cases.

---

> ### Author Rebuttal · Authors · 2024-08-05
>
> Dear Reviewer 5iMV,
>
> We are grateful for the time and effort you have invested in providing detailed and insightful feedback. We appreciate your recognition of the key contributions of our method, including eliminating the need for human annotation, the commendability of trajectories generated via MCTS, and proposing step-level beam search to reduce inference cost.
>
> Distinctions from other works can refer to `G1 in general rebuttal`. Then, please find our detailed responses to each of your points below.
>
> > W1: The paper's results are all program-aided, a fact not clearly identified in the text. Consequently, the comparison with non-program-aided LLMs may not be fair enough.
>
> In `Line 208-211` of Section Experiments, we mentioned that `To ensure a fair comparison, we primarily contrast our approach with the highest-performing SFT models that utilize an external tool - a Python code interpreter. These include MAmmoTH [43], MathCoder [35], ToRA [12], MARIO [44], MathGenie [23], and DeepSeek-Math-Instruct [29]. More implementation details can be found in Appendix C.`. In addition, in the `Tool column in Table 2`, we also marked whether each method uses program.
>
> Most of the 7B SOTA methods, e.g., ToRA, DeepseekMATH-instruct, use python programs to perform mathematical reasoning. As the program can mostly address the numerical computation in math reasoning. For larger LLMs, a recent example shows that most LLMs may fail for "9.9 and 9.11, which is bigger?". However, with the assistance of program, LLMs can easily solve such problems [r1].
>
> [r1] Solving Challenging Math Word Problems Using GPT-4 Code Interpreter with Code-based Self-Verification
>
> The comparison with non-program-aided LLMs is for reference only, which is a convention in previous works, e.g., ToRA, MAmmoTH, etc.
>
> Furthermore, previous works relied on human or GPT-4 annotation process to supervise the data, as shown in the **Seed Data Annotation** column of Table 2. Our **almost unsupervised/weakly supervised setting** v.s. **supervised setting** in previous works is actually an unfair comparison to our method, but we still achieve very competitive results.
>
> We hope this clarification addresses your concern, and we are open to further discussions should you have any additional questions.
>
> > W2: The benchmark only includes GPT models in the 'Proprietary Models' section. It would be beneficial to include other LLMs such as Gemini and Claude for a more robust evaluation.
>
> First, we include the GPT-4 results quoted from previous works for reference only, by following the convention of related works. The "Proprietary Models" are not fully accessible all over the world and require global credit card for the API usage. It is difficult to evaluate on other proprietary models, especially on the OOD data, but we would like to include the GSM8K/MATH results of Gemini and Claude from relevant papers in revised version.
>
> Additionally, all "Proprietary Models" should be larger in size than the 7B model, which is the main focus of our related works, making it difficult to justify fairness.
>
> > W3: An image illustrating the Step-Level Beam Search (SBS) would greatly aid in explaining the algorithm.
>
> In `Line 168-178` and `Algorithm 2`, we introduced the SBS by modifying MCTS. However, we can view the SBS from the perspective of traditional (token-level) beam search (BS).
>
> For simplicity, we use SBS(1, 5) and max step=3 for illustration and we will add an image in our revised paper.
> 1. The policy model samples 5 candidates for step 1, denoted as $a_1^1, a_1^2, \ldots, a_1^5$. (**similar to 5 candidate tokens in BS**)
> 2. The value model reranks 5 candidates, and selected the best, e.g., the top-1 is $a_1^1$. (**BS may use some variant of logprobs for reranking, such as length penalty version**)
> 3. The policy model samples 5 candidates for step 2 based on prefix $a_1^1$, denoted as $(a_1^1,a_2^1), (a_1^1,a_2^2), \ldots,( a_1^1,a_2^5)$.
> 4. The value model reranks 5 candidates of first two steps, and selected the best. For example, we assume the top-1 is $(a_1^1,a_2^1)$.
> 5. Repeating above process in 3 and 4 will get the final best path $(a_1^1, a_2^1, a_3^1)$.
>
> > L1: The framework does not generate a reward model that can facilitate other LLMs, which potentially limits its applicability.
>
> Thank you for your helpful comments. We would like to address that our framework can also facilitate other LLMs.
>   - The step-level evaluation signals (Q-values) generated by our framework can be used to train a separate value model. Reviewer 2HCk concerned about the performance of training the value model and policy model separately. We share the experimental results here.
>
> |$B_1=1$|GSM8K|MATH|GK2023|OCW|avg. time (s/q) & GPU usage|
> |--|:--:|:--:|:--:|:--:|:--:|
> |two separate models|82.4 |61.4|46.4|29.9|4.3 / $2\times$A100-80G|
> |AlphaMath (Ours) |81.1|62.8 |46.2|30.5| 3.1 / $1\times$A100-80G|
>
> From the table, our AlphaMath has similar overall performance to the two separate models, but two separate models requires more inference time and computing resources, which is what we want to avoid.
> - Our joint model (when using value model head only) can also score other LLMs with Python Code as long as the output format of other LLMs is reorganized by following the our defined REACT format.
> - If the LLMs insist to use CoT solution, it needs to retrain the value model in the MCTS framework with step-level CoT solutions.
>
> > L2: The framework cannot be extended to tasks with open-ended outputs, limiting its potential use cases.
>
> In `Line 518-520`, we have stated that our approach only works on tasks that are evaluated based on the actual answers in future work section. To further clarify, we will add the scope of our method in **Limitation**.
>
> We sincerely appreciate your invaluable feedback and would be delighted to engage in further discussion to address any questions you may have, with the aim of improving and enhancing the quality of our work.
>
> Sincerely,
>
> Authors of 6802

---

> > ### Comment · Reviewer_5iMV · 2024-08-08
> >
> > Thank you for your reply!
> >
> > I am curious about the OCW dataset you used, can you provide some information about it and maybe a link to its source?

---

> ### Author Response · Authors · 2024-08-08
>
> Dear Reviewer 5iMV,
>
> Thank you for your continued engagement.
>
>
> In `Lines 199-202 and 682-684` of our paper, we provide a detailed introduction to this dataset. Specifically, OCWCourses (OCW) focuses on STEM (Science, Technology, Engineering, and Mathematics) problems, particularly emphasizing calculations in physics, chemistry, and biology, which present a high-quality out-of-domain dataset and greater challenges compared to MATH. This dataset is sourced from [1]. The dataset  [**link**](https://huggingface.co/datasets/zhangirazerbayev/ocwcourses) on huggingface hub. As the DeepSeek uses this testset and one of our main models is built upon DeepSeek pretrained base LLM, we also use this testset for a fair comparison.
>
> We hope this clarification addresses your concerns.
>
> Additionally, do you still have questions about your previous concerns? We would appreciate it if you could share any further questions you may have. We look forward to the opportunity to enhance your understanding of our work through continued dialogue.
>
>
> [1] Lewkowycz, Aitor, et al. **Minerva: Solving quantitative reasoning problems with language models.** Advances in Neural Information Processing Systems 35 (2022): 3843-3857.
>
> Sincerely,
>
> Authors of 6802

---

> ### Comment · Area_Chair_UD49 · 2024-08-12
>
> Dear reviewer,
>
> Can you let the author and us know if you are satisfied or not with their rebuttal, and if you have any further comments?
>
> Thanks,
>
> AC

---

> > ### Comment · Reviewer_5iMV · 2024-08-12
> >
> > Dear AC and reviewer:
> >
> > Sorry for replying late, I have read the rebuttal and the comment. I will keep my rating. Thanks!

---

### Official Review · Reviewer_15WC · 2024-07-12

**Soundness:** 3
**Presentation:** 2
**Contribution:** 3
**Rating:** 4
**Confidence:** 5

**Summary:**

The authors propose the AlphaMath method, which leverages Monte Carlo Tree Search (MCTS) to improve the mathematical reasoning capabilities of large language models (LLMs) without requiring expensive human or GPT-4 annotations for process supervision. The framework integrates a value model with the LLM to generate and evaluate reasoning steps autonomously. Experimental results demonstrate that AlphaMath achieves comparable or superior performance to baseline methods on both in-domain and out-of-domain datasets.

To AC and the authors: The detailed comments are mainly covered in the **Questions** section. Please refer to the section for details. Generally speaking, I would consider the paper as very close to the threshold. If the authors can properly address the concerns in the below section, this could be a valuable study in this area.

**Strengths:**

MCTS is a powerful algorithm for complex planning. Mathematical reasoning is a very important but challenging area. Even the most advanced LLMs nowadays can easily make mistakes when solving a mathematical problem. There are many recent attempts using MCTS on mathematical tasks, and this paper is a good one among many. The empirical results demonstrate the effectiveness of the proposed method.

**Weaknesses:**

Many presentations in the paper are somewhat unclear, requiring further clarification and elaboration. There are also some minor over-claims in the introduction. There are many modifications comparing to the original AlphaGo and Zero paper but the authors typically do not provide explanations for such changes. Some tables and figures are confusing. Please see the **Questions** section for the details.

One thing I'd like to especially point out is, the title "almost zero" is kinda misleading. One would interpret it as using very little data, e.g., a few seed data. However, the method would still need the full training set of GSM8K/MATH.

**Questions:**

1. The authors use Table 1 and corresponding statements in the main body (line 25) to claim that *the annotation of mathematical solutions primarily relies on domain experts or GPT-4 in current efforts*. This claim is inaccurate, as multiple studies employ methods such as the Monte Carlo method [1,2] or even Monte Carlo Tree Search (MCTS) [3,4,5] to automatically annotate mathematical solutions. It is totally understandable that some related studies might have been missed, especially those posted after NeurIPS submission [3,5]. However, there are studies [1,4] published last year that should have been properly considered. I recommend that the authors revisit their claims and appropriately cite and discuss works that do *annotate mathematical solutions without human intervention or GPT-4*. This paper already presents numerous promising contributions, which I am very impressed; therefore, it is advisable to carefully avoid over-claiming in minor aspects that may negatively affect the first impression of readers and researchers familiar with this field.
2. In line 76, the authors write *the solution process can be broken into multiple reasoning steps (e.g., segmenting the solution based on distinct stages or simply on a period).* What exact step segmentation strategy do the authors use in this study?
3. In line 118, the authors write, *we employ sampling generation with higher temperature to ensure diversity* in generating actions for a step. This method requires further elaboration. From my understanding, it is non-trivial for an LLM to generate only a single mathematical step with early stopping. If the authors generate an entire solution and truncate it to retain only the subsequent step, this approach would be computationally expensive.
4. In line 125, the authors write *we follow a trade-off rollout strategy between AlphaGo and AlphaGo Zero. Because our tree depth is much shallower than Go games (e.g., a maximum depth of 8).* What is the "trade-off rollout strategy between AlphaGo and AlphaGo Zero"? I don't personally remember such a concept in the AlphaGo work series; and why your tree has a maximum depth of 8? There should have been many solutions with more steps for the MATH dataset.
5. In line 135, *assuming that ...*. Why this assumption is valid? Please elaborate.
6. In Eq. (6), the the loss terms for the value model is minimizing the difference between Q and V. However, in AlphaGo Zero it is minimizing the difference between V and the final reward z (1/-1 indicating the final winner at the termination state). The authors do not provide an explanation of this modification. Please elaborate.
7. In line 164, *This selection is guided by the maximum Q-value*. In AlphaGo and AlphaGo Zero, the selection at inference time is guided by the visited count. Using Q is also a valid choice, but there should be clarification for this modification. An optional ablation study would be better.
8. In Table 3, the SBS with $B_1=2 \text{or} 3$ has shorter average time than $B_1=1$. This looks weird. I guess it comes from the less number of steps. But why $B_1=2 \text{or} 3$ has less number of steps then? Also, the "discussion of majority voting" is really confusing. I still don't understand why they are not comparable after reading it.
9. In Figure 5, where is the Q-value sampled from? For full solution only or also including intermediate of steps? Also why there are three curves in the right figure while there are only two legends?

Ref:

1. Math-Shepherd: Verify and Reinforce LLMs Step-by-step without Human Annotations. https://arxiv.org/abs/2312.08935
2. Multi-step Problem Solving Through a Verifier: An Empirical Analysis on Model-induced Process Supervision. https://arxiv.org/abs/2402.02658
3. Improve Mathematical Reasoning in Language Models by Automated Process Supervision. https://arxiv.org/abs/2406.06592
4. Alphazero-like Tree-Search can Guide Large Language Model Decoding and Training. https://arxiv.org/abs/2309.17179
5. Step-level Value Preference Optimization for Mathematical Reasoning. https://arxiv.org/abs/2406.10858

**Limitations:**

The authors provide a limitations section (A.1); however, it omits several critical aspects:

1. Methods that use the final answer's correctness to infer the reward or value of intermediate steps are prone to false positive and negative issues. Specifically, when the final answer is correct, all intermediate steps are rewarded to some extent. Nevertheless, false positive cases occur when models luckily guess the correct final answer despite incorrect reasoning. Previous related studies, such as Math-Shepherd and OmegaPRM, have acknowledged this limitation. It is recommended that this paper also addresses it in the limitations section.
2. The method relies on an automatic verifier to determine the correctness of the final answer. This verifier is not always applicable to many tasks, such as open-ended text generation (e.g., translation and summarization). The authors did not clearly explain the limitations concerning the scope of tasks.

---

> ### Author Rebuttal · Authors · 2024-08-06
>
> Dear Reviewer 15WC,
>
> We sincerely appreciate your efforts in evaluating our manuscript. We have carefully considered your insights and critiques, then address each of your comments.
>
> > W1: Distinctions to [1,4]
>
> Thanks for raising this issue. We will avoid over-claiming in future version. Highlighted contributions and distinctions to [1,4] can refer to the `G1 in General Rebuttal`.
>
> > W2/W3: Step segmentation; How to generate/stop a step
>
> W2 and W3 are relevant, so we will address together.
>
> In `Lines 190-192`, we already mentioned that the step is defined in the form of **LLM REACT** format and provided the examples in `Appendix F`. The REACT format step includes Thought(text analysis) / Action(python interpreter) / Action input(code snippet) / Observation(code output). In `Line 620-627`, detailed definitions and examples of the $C$-steps and $A$-steps are also provided. $C$-step is REACT step including code, and $A$-step is the last REACT step including final summary only.
>
> We follow the REACT format to generate and stop a single step. In REACT, one can define the possible stop tokens for Observation. As shown in Appendix F, we can define the stop tokens, as "Observation" for round 0 and ["</code>", "</step>"] after round 0. When encountering any stop token, the LLM generation will stop. In **open-source toolkit vllm or Openai API**, stop token is an argument, thus it's not necessary to generate the entire solution and truncate it. In summary, the step-level sampling can still be efficient.
>
> More details can refer to REACT paper or relevant API document. This is not our focus, but we will add a relevant description in future version.
>
> > W4: Explain trade-off rollout and max depth=8
>
> We apologize for misuse of **follow** in line 125, which should be **design**. In AlphaGo, every rollout requires a light-weighted fast policy to sample hundreds steps in depth until terminal node. In AlphaGo Zero, there is no rollout at all, and the value estimation only depends on prediction of value model in previous round. Our defined value estimation $(1-\lambda)V_{\phi}(s_t) + \lambda r(s_{t+1})$ has the same form as AlphaGo. Differently, we set $\lambda$ as indicator function $I_{terminal}(s)$. `Lines 126-128` also explain it means such setting rollouts for only one-step. If the rollout reaches terminal, value adopts the only reward, otherwise, the value model prediction.
>
> As we explained for REACT, depth=8 means 7 code steps + 1 summary step. Even for MATH, 7 code snippets are enough to solve most problems. E.g., in ToRA (ref [12] in paper) found GPT-4 can solve 83.1% MATH problems within 3 code snippets. Since our base model is weaker than GPT-4, we relax it to 7.  **Table 4 in PDF file** shows the detailed step statistics.
>
> > W5: Explain assumption
>
> Our assumption is `$Q(s_t, a_t) = r(s_t, a_t) + V(s_{t+1}) = V(s_{t+1})$ for non-terminal nodes`.
> - The 1st "=" is not assumption, but the definition of state-action and state values [r1,r2]. It is due to deterministic transition $s_{t+1}=cat(s_t, a_t)$ in Eq. (1) for LLM output.
> -  We mentioned why the 2nd "=" holds in `footnote 1`. The reward of non-terminal nodes is 0. This is a common definition in RL if the reward can only be determined until the end of the episode.
>
> [r1] Sutton & Barto's Reinforcement Learning: An Introduction
>
> [r2] The UC Berkeley course: Deep Reinforcement Learning (CS285)
>
> > W6: Explain labels are V, not z
>
> In standard RL, most value fitting algorithms aim to learn intermediate or step-level values [r1,r2] with various value estimation techniques. Therefore, we didn't make the modification; instead, AlphaGo series did.
>
> The reason why learning from final reward works is that AlphaGo series applied thousands of simulations or sampled trees in MCTS. In the MSE value fitting, which is theoretically a regression problem, the regression signals 1 and −1 are proportional to the number of positive and negative rewards in simulations. The ratio asymptotically approaches the true value due to the large sampling size. Thus, the overall gradient direction for MSE should be asymptotically accurate. In LLMs, we cannot afford thousands of simulations.
>
> > W7: Explain value selection in MCTS inference
>
> Selection with maximum Q-value is not a modification, but one of popular methods, as introduced by textbook [r1] Sec. 8.11 MCTS, quoted as `some mechanism that depends on the accumulated statistics in the tree, e.g., the largest action value, or action with the largest visit count)`.
>
> Similarly, due to the large amount of simulations in AlphaGo series, using visit count is a robust estimation. In our case with much fewer simulations, we found maximum Q-value is more robust. In the submitted code **offline_inference.py**, we already implemented visit count. For the response purpose, we include the relevant result in **Table 5 of PDF file**.
>
> Actually, using the variant of visit count $\propto N^\alpha$ in AlphaGo series is a modification. Previous works such as [4] follow the variant without considering sampling size, which may lead to suboptimal.
>
> > W8: Discuss large B1 with fewer steps, and majority voting
>
> As we explained in previous response, each REACT code step includes text analysis, a code snippet and its execution output. If the output indicates an error message, the LLM will generate a new REACT step by rewriting the code snippet until no error occurs or max depth reaches. When B1 increases, there are more candidates for the value model to select from. Our MCTS training framework encourages the value model to choose reasoning steps that don't output error message. Thus, the average number of overall reasoning steps decreases as B1 increases.
>
> Please kindly refer to `G2 in General Rebuttal` for the discussion of majority voting.
>
> > W9 and Limitations
>
> Please kindly refer to `G3 in General Rebuttal`.
>
> We sincerely appreciate your valuable feedback and look forward to address any further concerns you may have.
>
> Sincerely,
>
> Authors of 6802

---

> > ### Author Response · Authors · 2024-08-10
> >
> > Dear Reviewer 15WC,
> >
> > Although we have made responses to your W1 in `G1 of general rebuttal`, we would like to clarify again the difference between our work and the two works[1,4] you mentioned.
> >
> > For **Alphazero-like[4]**, the training of policy and value models is entirely independent of MCTS. As shown in the appendix E.2 of [4], they collect multiple paths through rejection sampling, and train the value model through Monte Carlo (MC) estimate, TD-$\lambda$, or PPO. As mentioned in `G1 of general rebuttal`, their training of value model has nothing to do with MCTS. In contrast, our training falls within the true AlphaZero framework (MCTS + no annotation). As mentioned in your several questions (W6, W7), compared to AlphaZero, we have introduced some modifications to enhance the stability of AlphaMath's training. It is important to note that these adjustments are not modifications in the sense of traditional RL, but rather commonly used strategies within reinforcement learning (RL) and MCTS [r1,r2].
> >
> > So the core of **Alphazero-like[4]** is to propose an independent value model to help the policy model perform better reasoning through MCTS inference, arather than training. Moreover, their MCTS inference is very inefficient. As shown in `Lines 168-172` of our paper, we have modified the MCTS inference and proposed a more efficient step-level beam search algorithm. In addition, our value model and policy model share most parameters, which significantly reduces training and inference costs.
> >
> > For **Math-Shepherd[1]**, this work is even more remote from our work. The core of this work is to train a value model for reranking output. The form of collecting training step-level value siginals is Monte Carlo (MC) estimate, which is very inefficient and has high variance. They may also apply PPO for training, which is another far different RL algorithm with MCTS.
> >
> > In summary, Math-Shepherd and Alphazero-like are similar works. They aim to train a value model to help the policy model to better reason (Verification ranking for Math-Shepherd, MCTS inference for Alphazero-like). However, the learning of step-level value function through MC estimate, TD-$\lambda$ or PPO, which is fundamentally different from our work.
> >
> > We hope this clarification alleviates any concerns you may have had, and we look forward to engaging in further constructive discussion with you.
> >
> > Sincerely,
> >
> > Authors of 6802

---

> > ### Comment · Reviewer_15WC · 2024-08-12
> >
> > Thank author for the rebuttal. I've read through all the comment. I will continue discuss with AC and the other reviewers for the final decision. There's no further question for now.

---

> ### Comment · Area_Chair_UD49 · 2024-08-12
>
> Dear author,
>
> Can you let the author and us know if you've read the rebuttal, and if you have any further comments?
>
> Thanks,
>
> AC

---

### Official Review · Reviewer_otxC · 2024-07-12

**Soundness:** 3
**Presentation:** 2
**Contribution:** 2
**Rating:** 5
**Confidence:** 3

**Summary:**

This paper leverages the Monte Carlo Tree Search (MCTS) to iteratively train policy and value models by automatically generating process supervision and step-level evaluation signals, eliminating the need for human-annotated process supervision data. Specifically, the method combines the inner capabilities of pre-trained LLMs with the MCTS framework to generate correct and incorrect solution paths and optimize the models based on the node evaluations along these paths. To improve inference efficiency, the paper also proposes a step-level beam search strategy that allows the value model to assist the policy model in navigating more effective reasoning paths rather than relying solely on prior probabilities while avoiding excessive time consumption. Experimental results demonstrate that even without GPT-4 or human-annotated process supervision, this paper' s method performs comparably or better than existing SOTA methods across multiple in-domain and out-domain datasets.

**Strengths:**

1. The work proposes a novel and efficient method that eliminates the need for human process annotations by leveraging the MCTS framework for model self-evolution.
2. The step-level beam search strategy significantly enhances the model's reasoning capabilities while maintaining computational efficiency.

**Weaknesses:**

1. Although the method does not rely on human-annotated process data, it still requires actual answers as reward signals. Future work could explore fully unsupervised approaches.
2. Although the step-level beam search improves efficiency, the MCTS method still has some limitations in terms of computational complexity, which makes its practicality indistinguishable from majority voting.

**Questions:**

N/A

---

> ### Author Rebuttal · Authors · 2024-08-05
>
> Dear Reviewer otxC,
>
> We greatly appreciate your valuable comments and are pleased that you recognize the novelty and efficiency of our proposed method. Below, we address your comments in detail.
>
> > W1: Although the method does not rely on human-annotated process data, it still requires actual answers as reward signals. Future work could explore fully unsupervised approaches.
>
> Thanks for the insightful suggestion. In `Line 499-505` of Section Future Work, we also mentioned the possibility of "Really from Zero", quoted as `A challenging yet profoundly meaningful future direction is to identify an appropriate reward definition that allows AlphaMath to eliminate dependence on actual answers, thereby achieving really from zero. `.
>
> A promising trial can be exploring in proof problems by leveraging the **Lean** programming language. Unlike traditional problems that require a final answer, proof problems focus on verifying both numerical computations and proof correctness. The Lean language is well-suited for this purpose, especially for proof derivation. For example, recent AlphaProof [r1] achieved 2024 IMO silver medal by applying our method-alike approach with Lean.
>
> [r1] DeepMind. AI achieves silver-medal standard solving International Mathematical Olympiad problems
>
> > W2: Although the step-level beam search improves efficiency, the MCTS method still has some limitations in terms of computational complexity, which makes its practicality indistinguishable from majority voting.
>
> Thanks for pointing out this issue. We totally agree with your claim that compared with majoring voting, inference with MCTS requires larger computational complexity. This is also exactly the reason why we mainly use MCTS in training rather than inference. Value estimation with MCTS can provide more accurate learning signals than simple Monte Carlo estimate, so training with MCTS framework is most beneficial to value model.
>
> To address the inference issue of MCTS, we further proposed to modify it and achieve a very efficient inference method--step-level beam search (SBS, note that SBS cannot be used for training). In `Line 168-172`, we  mentioned that `However, MCTS is computationally intensive for simulations, making it less viable for use in production environments. To address this, we modify the MCTS inference process by eliminating the backup operation, introducing a simplified method, which we refer to as Step-level Beam Search (SBS), detailed in Algorithm 2. This approach does not construct the entire tree; instead, it dynamically selects the best child node during node expansion.`
>
> In `Tables 2 and 3` of our paper, we also demonstrate the significant advantages of SBS in efficiency and effectiveness compared to MCTS inference, greedy decoding or majority voting. For example, SBS ($B_1=3$) is 5 times faster than MCTS and has 1.7% better performance. Meanwhile, the efficiency of SBS is similar to that of greedy decoding, with only a difference of 0.8s per question, but the efficiency is improved by 12.1% on the MATH dataset. We believe that these aspects demonstrate the superiority of our proposed SBS.
>
> Compared to majoring voting, `Table 3` of our paper cannot fully show the efficiency difference between SBS and maj@n. However, if we scale $n$ to large numbers, from the following table, we can see SBS shows significant advantage on accuracy.
>
> |n|Maj@n|Avg time s/q|SBS ($B_1=n$)|Avg time s/q|
> |--|:--:|:--:|:--:|:--:|
> |1|||62.32||
> |2|||64.66||
> |3|56.82||65.74|2.3|
> |5|61.84|2.9|65.98|4.7|
> |10|62.58||||
> |15|63.56||||
> |20|64.16|15.9|||
>
> We hope the above clarifications have enhanced your understanding of our work. We appreciate your valuable feedback and are eager to further discuss any additional questions or comments you may have to improve our manuscript.
>
> Your insights are invaluable in our endeavor to refine and strengthen our research. Thank you once again for your thoughtful review.
>
> Sincerely,
>
> Authors of 6802

---

> > ### Comment · Reviewer_otxC · 2024-08-13
> > **Remain the rating**
> >
> > I've read the authors' responses and am satisfied with the responses to my questions. I've also read the comments from other reviewers. Based on all these, I decide to keep the rating.

---

> ### Comment · Area_Chair_UD49 · 2024-08-12
>
> Dear reviewer,
>
> Can you let the author and us know if you've read the rebuttal, and if you have any further comments?
>
> Thanks,
>
> AC

---

### Official Review · Reviewer_EVyZ · 2024-07-14

**Soundness:** 3
**Presentation:** 3
**Contribution:** 3
**Rating:** 5
**Confidence:** 4

**Summary:**

The paper introduces a novel approach leveraging the Monte Carlo Tree Search (MCTS) framework to generate process supervision and step-level evaluation signals automatically, thus enhancing the mathematical reasoning capabilities of LLMs. This method bypasses the need for costly and labor-intensive manual annotations by allowing the models to train iteratively on both policy and value aspects, with an efficient inference strategy called step-level beam search. This approach demonstrates significant improvements on various datasets, showing comparable or even superior results to existing state-of-the-art methods without relying on external process annotations.

**Strengths:**

The paper leverages a Monte Carlo Tree Search (MCTS) framework to automatically generate both the process supervision and step-level evaluation signals necessary for training LLMs. This approach avoids reliance on human annotated data by generating necessary training signals through interactions within the MCTS environment. As a result, the model achieves comparable or even superior performance to previous state-of-the-art methods on both in-domain and out-of-domain datasets.

**Weaknesses:**

1. Duplicate citations, such as [6], [7]
2. Appendix Figure 9, the reason why the performance of the third round model in MATH at Level 1 and Level 3, Counting & Probability, Number Theory and other categories has declined. Whether it will continue to decline after more rounds.
3. The method integrates CoT and Code for cross-use. How does the performance of the model change if only CoT is used?
4. Evaluate on more out of domain benchmarks, such as GSM8k_hard and GSM8k_robust and OpenLLM Leaderboard
5. How much total training data is used. Explore the effect on other base model, such as Llama2, Mistral
6. Explore the effect of different values of the hyperparameter β in Eq. (8).
7. what is the change in loss of Policy model and Value model during training.
8. What's the accurate of the Value model predictions based on the current State and Action.
9. Insufficient introduction of related work, such as Mathematical Reasoning and MCTS.
10. Differences reasoning ability and solution generation process across different rounds for the policy model , as well as the scoring differences by the Reward Model.
11. How the ground truth label scores for training the first round of the Value Model are obtained, and the procedures for the second and third rounds.

**Questions:**

No question besides the one raised in weakness.

**Limitations:**

Yes

---

> ### Author Rebuttal · Authors · 2024-08-02
>
> Dear Reviewer EVyZ,
>
> We greatly appreciate your detailed review and the recognition of the strengths of our paper. Due to word limit, we apologize for shortening your questions. We will address each of your comments in detail, hoping to alleviate any concerns you may have.
>
> > W1: Duplicate citations
>
> We apologize for the error and will correct all reference errors in the revised version.
>
> > W2: Explain acc of some levels in round 3 declined
>
> Thanks for highlighting this issue. In `Line 194-195`, we have discussed the selection of $K$ using the **Elbow method** on overall accuracy. The details including next round (round 4) can refer to **Table 1 in PDF file**. As we discussed in `Line 564-568`, the mixed increase and decrease performance on finer-grained levels is normal after training (overall accuracy) converges.
>
> > W3: What about CoT
>
> Due to limited time, we cannot do full experiments on CoT. For a fair comparison, we follow most SOTA methods in the setup of CoT+Code, because it has been proven (eg, [r1] or ToRA) that the performance of COT is less effective to CoT+Code, and LLMs struggle with numerical calculation. A recent famous example is that LLMs fail to compare 9.9 and 9.11.
>
> > W4: Other OOD data
>
> Thanks for the suggestion. We did not use the GSM8k_hard dataset because it does not represent a truly OOD scenario. It uses the same questions of GSM8k by varying digits, and many factual errors exists, such as the negative number of people.
>
> As introduced in Appendix C.5, the two OOD datasets we used are significantly more challenging than GSM8K and MATH, with different question sources and domains (e.g., physics). So they can better represent the true OOD scenario.
>
> For the response purpose, we evaluate on GSM8K-robust and its corresponding method, as shown in **Table 2 in PDF file**.
>
> > W5: Training data size; Try Llama2 or Mistral
>
> In `Line 186-187` , we noted that our raw training data only contain **15K QA pairs** from GSM8K/MATH. Using these 15k pairs, we generate **59k solution intermediate processes** by MCTS, significantly smaller than existing SOTA (**776K** in DeepseekMath-instruct). In addition, process annotation does not rely on GPT-4 or human.
>
> In `Table 4` of Section 4.6 and `Table 5` in Appendix, we already explored the effect of other base LLMs, including general LLM **Llama3**, and SFT models (MARIO). We think Llama3 can represent the general LLMs, such as Llama2 and mistral.
>
> > W6: Hyperparameter beta in Eq(8)
>
> We assume it refers to Eq(6). We conducted hyperparameter selection in our preliminary experiments. In this response, we would like to share our findings that $\beta$ does not significantly impact model performance.  We list the results in **Table 3 in PDF file** for your reference.
>
> > W7: Log of training loss
>
> We plot the loss in **Figure 1 in PDF file** for your reference.
>
> > W8: Accuracy of value model
>
> Thanks for your valuable comments. It is theorectially difficult to evaluate accuracy of intermediate step values without human annotations. Even in the realm of reinforcement learning(RL), it is also a difficult task if no step reward is returned. Thus, most RL  algorithms care more about the values ranking (e.g., via argmax), given current state or action.
>
> In `Figure 5` of Section 4.5, we have followed the convention of RL to illustrate the relationship between the value predictions of intermediate and final steps by plotting the distribution of $Q$-values against final reward/correctness. It offers a more intuitive understanding. On the testset, our value model predictions align well with expectations, particularly for correct steps.
>
> In `Line 277-298`, we also discussed the minor flaws of the value model. As shown in Fig. 5, the value model will score $\approx1$ for some final wrong solutions, because the final incorrectness doesn't mean all steps are wrong, and correct steps may exist for early timestamp.
>
> > W9: More related works
>
> Due to space limitation, we focused on data annotation and value model, as our title indicated. We totally agree that other fields, such as MCTS, are also important and we will revise accordingly.
>
> > W10: Performance of different rounds
>
> Thanks for your insightful comments. Actually, in `Figure 3/4/6/8/9, Table 5, Appendix D`, we have performed extensive experiments to investigate the performance of different rounds, that covers your concern.
> - policy model in Figure 3/4/6/8/9, greedy decoding w.o. value
> - value model in Figure 4/6, SBS inference with value
> - case studies in Appendix D, intuitively show how Q-values changes
> - problem solving rate in Figure 3/9
>
> > W11: Labels of value model
>
> We aim to clarify it by discussing the offline Q-learning method. Generally, the initial $Q$ values of each node in the MCTS tree are set to 0 and are subsequently updated as indicated in `Eq. (4) and $Q$ update in Line 132`.
>
> `Lines 143-147` explain how the values are calculated for the 1st round. Since the value model is randomly initialized, the value head's prediction (1st rightside term in Eq. (4)) should be $\approx0$. Thus, $Q$ update is primarily influenced by the final reward/correctness. Theoretically, with more MCTS simulations/rollouts, the $Q$ values will gradually converge to their true underlying ground truth labels.
>
> For other rounds, `Lines 120-132` describe the value calculation process. In 1st rightside term of Eq. (4), the value model is trained on the previously collected values, providing a more accurate prediction than $\approx0$ in 1st round. Thus, the $Q$ update in Line 132 may rely not only on the final reward but also on the value model's prediction.
>
> [r1] Solving Challenging Math Word Problems Using GPT-4 Code Interpreter with Code-based Self-Verification
>
> We believe our explanations have clarified your concerns and offer a clearer understanding of our work. Thank you for your constructive feedback. We are eager to engage in further discussions to enhance and refine our work.
>
> Sincerely,
>
> Authors of 6802

---

> ### Comment · Area_Chair_UD49 · 2024-08-12
>
> Dear reviewer,
>
> Can you let the author and us know if you've read the rebuttal, and if you have any further comments?
>
> Thanks,
>
> AC

---

> > ### Comment · Reviewer_EVyZ · 2024-08-14
> >
> > I have read the authors' response. It addressed some of my concerns. So I will maintain my current score.

---

### Official Review · Reviewer_2HCk · 2024-07-16

**Soundness:** 3
**Presentation:** 3
**Contribution:** 3
**Rating:** 5
**Confidence:** 3

**Summary:**

This paper utilizes Monte Carlo Tree Search (MCTS) to sample high-quality process-supervised data for iterative training, effectively reducing the dependency on GPT-4 and thus lowering the associated costs. Additionally, the authors propose an inference strategy, step-level beam search, which leverages value models to enhance the efficiency and effectiveness of the tree search process.

**Strengths:**

1. This paper is well-written, with a clear introduction to MCTS and the authors' proposed method, step-level beam search (SBS).
2. Their results effectively demonstrate the superiority of SBS over MCTS.
3. The model they trained shows a significant improvement over the baseline model, with an increase in performance from 33.2 to 53.6.

**Weaknesses:**

1. The comparison in the experimental results appears to be rather unfair. Generally, more computation leads to higher performance. In Table 2, the authors compare their fine-tuned model using SBS with greedy decoding for other open-sourced models. In Table 3, they compare SBS with majority voting. It is possible that the improvement stems from the reward model rather than the SBS algorithm itself. The authors should also report the results of best-of-n or weighted majority voting using their value model to demonstrate the effectiveness of SBS.
2. The experimental results are not detailed enough in Tables 2 and 3. The authors should also include major@n with different n values. Perhaps they can plot figures with $B_1$ and $n$ as the x-axis to provide a more comprehensive view.

**Questions:**

1. As mentioned in the weaknesses, what are the results for best-of-n or weighted voting? Additionally, how does the accuracy change as n increases during majority voting?
2. How will the performance change when separating the policy model and value model as two totally different models?

**Limitations:**

Yes

---

> ### Author Rebuttal · Authors · 2024-08-05
>
> Dear Reviewer 2HCk
>
> We sincerely thank you for your thorough evaluation and for acknowledging the clarity of our paper, as well as the effectiveness and significant performance improvements we've demonstrated. Due to word limit, we apologize for omitting your question and replacing them with numbers and key words. Below, we address each of your comments in detail.
>
> > W1: Unfair comparison. Relationship between reward model and SBS, try best-of-n.
>
> We appreciate the reviewers' feedback and the opportunity to clarify experimental comparisons.
>
> 1. In `Line 223-224`, we mentioned the greedy decoding results are included in Table 2. To clarify, it is marked as **AlphaMath (K=3)**, where $K$ means the training iterations. Actually, due to our novel experimental setups, the experimental comparison puts our results at a disadvantage relative to previous works.
>
>   - Our training method is categorized as **almost unsupervised / weakly-supervised** learning, whereas previous works primarily employ **supervised learning**. If we were to exclude the human or GPT-4 annotated solution processes from the training data in previous works, and train their models using only question-answer pairs, we hypothesize that their performance would drop drastically.
>   - The dataset for supervised training in previous SOTA DeepSeek-Math-Instuct includes **776K question-process-answer** tuples. In contrast, our method only relies on the **15K question-answer** pairs.
>   - As shown in Table 2, in greedy decoding setup, AlphaMath achieves a performance score of 53.6, which is comparable or better than all previous supervised methods except DeepSeek-Math-Instuct's 57.2. The relatively small gap between an **almost unsupervised** method and a **supervised** method is noteworthy and underscores the potential of our approach.
>
> 2. As illustrated by the SBS (step-level beam search) inference in Algorithm 2, our results are already presented as **best-of-n**, but on a step-level basis. The value loss (the 2nd and 3rd rightside terms in Eq. (6)) indicates that the value model is trained at the step-level. During inference, for example, in a setup of $B_1/B_2=1/5$ the SBS needs to choose the best out of 5 generated outputs at each step, using the step-level score prediction provided by the value model. In summary, the value model and SBS inference algorithm are highly integrated to improve the performance, by selecting step-level **best-of-n**.
>
> > W2: More n for maj@n.
>
> Thanks for the suggestions to make our experiments in a more comprehensive view.
>
> First, We want to clarify that majority voting is not the focus of our work. Our training algorithm is fundamentally different from previous methods, which naturally leads to a distinct SBS inference algorithm compared to the majority voting used in earlier approaches. We propose an autonomous step-level process annotation method by leveraging the cooperation between policy and value models within the framework of MCTS. This policy-value reinforcement learning framework naturally favors the SBS inference algorithm.
>
> Second, Table 2 is primarily used to compare our **almost unsupervised / weakly-supervised** results with previous methods in traditional **supervised learning**. Table 3 presents an analysis experiment of our own model to explore the trade-off between performance and efficiency using different inference algorithms.
>
> Third, as suggested, we would like to include the performance of maj@n with varying n values. From the table, our SBS is significantly better than major@n. More ressults can refer to `Table 6 in PDF file`.
>
> |n|Maj@n|Avg time s/q|SBS ($B_1=n$)|Avg time s/q|
> |--|:--:|:--:|:--:|:--:|
> |1|||62.32||
> |2|||64.66||
> |3|56.82||65.74|2.3|
> |5|61.84|2.9|65.98|4.7|
> |10|62.58||||
> |15|63.56||||
> |20|64.16|15.9|||
>
> We hope the above explanations address your concerns and help you better understand our work.
> > Q1
>
> We have addressed the relevant issue in the response for Weakness 1 and 2.
>
> > Q2: Performance of separating policy and value models.
>
> In the realm of reinforcement learning (RL), whether or how to share networks between policy and value models is a long-debated design issue, [r1,r2] have noted that using two separate networks is usually easier to implement and more stable to train. However, this design is inefficient optimization as it doesn't allow for the sharing of internal representations. On the other hand, shared networks are challenging to train because the different losses generate gradients in various directions and scales, requiring careful tuning of the loss weights.
>
> Making the training of shared networks stable is actually one of the contributions in our work as a RL method. Beyond this consideration, our insistence on tuning the shared architecture is driven by computational cost. Using two LLMs during inference nearly doubles the computational expense. Sharing networks had also been adopted in some famous RL works, such as AlphaZero.
>
> For the response purpose, we'd like to include the new experimental results with two separate models. We set the temperature to 0.6, $B_1/B_2=1/5$, and compare our AlphaMath (joint training) with two separate models, as shown in the following table:
>
> |$B_1=1$|GSM8K|MATH|GK2023|OCW|avg. time (s/q) & GPU usage|
> |--|:--:|:--:|:--:|:--:|:--:|
> |two separate models|82.4|61.4|46.4|29.9|4.3 / $2\times$A100-80G|
> |AlphaMath (Ours)|81.1|62.8|46.2|30.5|3.1 / $1\times$A100-80G|
>
> From the table, our AlphaMath has similar overall performance to the two separate models, but two separate models requires more inference time and computing resources, which is what we want to avoid.
>
> [r1] Sutton & Barto's Reinforcement Learning: An Introduction
>
> [r2] The UC Berkeley course: Deep Reinforcement Learning (CS285)
>
> We hope the above clarifications will help you better understand our work. We are happy to communicate with you further if you have any questions to enhance our manuscript.
>
> Sincerely,
>
> Authors of 6802

---

> > ### Comment · Reviewer_2HCk · 2024-08-13
> >
> > Thank you for you rebuttal.
> >
> > > Our training method is categorized as almost unsupervised / weakly-supervised learning
> >
> > It doesn't seem right to claim training on question-answer pairs as unsupervised or weakly-supervised learning, though it's a good setting for bootstrapping reasoning [1].
> >
> > > The relatively small gap between an almost unsupervised method and a supervised method is noteworthy and underscores the potential of our approach.
> >
> > DeepSeek-Math base model is already trained on some instruction following data and achives 66.9 GSM8K and 31.4 MATH simply by prompting. But I agree that achiving 73.5 GSM8K and 53.6 MATH after RL training on 15K question-answer pairs is still impressive.
> >
> > > More n for maj@n.
> >
> > Thank you for the results. I'm surprised that your $n=20$ majority voting run takes $5\times$ longer than the $n=5$ run. I would have expected the $n=20$ run to take only $2-3\times$ longer due to GPU parallelism. Could the authors provide more details on the experimental settings?
> >
> > > Q1. We have addressed the relevant issue in the response for Weakness 1 and 2.
> >
> > What are the results of weighted voting and best-of-n? I believe a fair baseline should also incorporate the value/reward model during inference.
> >
> > > Q2: Performance of separating policy and value models.
> >
> > Thanks for the results.
> >
> > [1] Zelikman, Eric, et al. "Star: Bootstrapping reasoning with reasoning." Advances in Neural Information Processing Systems 35 (2022): 15476-15488.

---

> ### Comment · Area_Chair_UD49 · 2024-08-12
>
> Dear reviewer,
>
> Can you let the author and us know if you've read the rebuttal, and if you have any further comments?
>
> AC

---

> ### Author Response · Authors · 2024-08-13
>
> Thanks for your valuable response.
>
> > Our training method is categorized as almost unsupervised / weakly-supervised learning. The relatively small gap between an almost unsupervised method and a supervised method is noteworthy and underscores the potential of our approach.
>
> Thanks for your suggestion about [1]. We will cite this paper in our related works, and categorize our work as bootstrapping reasoning.
>
> > More n for maj@n. Could the authors provide more details on the experimental settings?
>
> We directly use the open-sourced `vllm` inference framework by setting the number of generations `n=20`. The parallelism is implemented by the vllm. We think the reason why the inference time of `n=20` is $5\times$ longer than `n=5` is that the Python code execution is on CPU rather than GPU. As we mentioned, we use the LLM REACT agent, including both text+code generation (GPU parallel) and code execution (CPU parallel).
>
> > What are the results of weighted voting and best-of-n? I believe a fair baseline should also incorporate the value/reward model during inference.
>
> In response of W1, we mentioned that as illustrated by the SBS (step-level beam search) inference in Algorithm 2, our results are already presented as best-of-n, but on a step-level.
>
> The value loss (the 2nd and 3rd rightside terms in Eq. (6)) indicates that the value model is trained at the step-level. During inference, for example, in a setup of
>  the SBS needs to choose the best out of 5 generated outputs at each step, using the step-level score prediction provided by the value model. In summary, the value model and SBS inference algorithm are highly integrated to improve the performance, by selecting step-level best-of-n.
>
> MCTS is a baseline that has already incorporate the value model.
>
> Since our **value model is step-level** in the setup of LLM REACT agent, for a fair comparison with incorporating the value model, we just implemented the step-level weighted voting based on our submitted code. For each step, we first sample 5 candidates, and select the step with largest weight summation. Then, we proceed to the next step.
>
> |method|Accuracy on MATH|
> |--|:--:|
> |step-level best-of-5, i.e. SBS(1,5)|62.32|
> |step-level weighted voting of 5, variant of SBS(1, 5)|61.94 (run 1), 62.36 (run 2)|
>
> Given our step-level value model, there is not significant difference of above two setups, demonstrating the robustness of our step-level value model and SBS inference algorithm.

---

### Author Rebuttal · Authors · 2024-08-06

> G1: Why our value model training works better?

We agree with previous works [1,4] (by Reviewer 15WC) may learn a reasonable policy model if they applied some autonomous annotation method, even they actually didn't do it. However, we respectfully argue that the value model learning in [1,4] may not favor learning from almost zero, as they require large sampling size or ORM initialization.

Our training of value model and those in [1,4] are **fundamentally different**, because we coherently integrated MCTS in both training and inference stages inspired by AlphaZero. This is the main reason why our model can successfully learn from almost zero without human annotations. Previous works utilized other reinforcement learning (RL) algorithms to learn the value model, even MCTS is used [4].

- **Value Estimation (most important)** As mentioned in `Line 91`, we didn't use Monte Carlo (MC) estimate [1,4] for value labels. MC estimate is an unbiased but high variance estimator. To fix it, variance reduction tricks are needed [r1,r2], e.g., increased sampling size, that may not work for high-dimension. Instead, we employ MCTS to construct value training labels. Theoretically, unlike MC estimate, which applies standard sampling over actions, the MCTS value estimate enables more implicit exploration. During tree expansion, MCTS prioritizes and selects non-greedy actions that may lead to better future returns. From the perspective of both practice and theory, MC and MCTS for value estimates are fundamentally different.
- **Process Supervision SFT Data** We did not utilize SFT data as [1,4]. Previous works employ other RL algorithms (e.g., PPO in [1]) that require initializing the policy model with a reference policy, typically an SFT model. Our training process adheres strictly to the AlphaZero framework, i.e., without human annotated process.
- **ORM Initialization** We did not use ORM to initialize value model (TD-$\lambda$[4] or MC estimate [1,4]). As [r1,r2] indicates, TD-$\lambda$ will introduce bias in the estimate. Tricks are needed to make the training less overfitting or degeneration. In our experience, we found a separate value model initialized from a SFT policy model can achieve more stable training and more robust predictions. Thus, we hypothesize that the ORM initialization is crucial in [1,4].
- **Network Sharing** In RL realm, it is well known [r1,r2] that two separate networks is easier to implement and more stable for training. This approach is inefficient data usage for optimization as it fails to share the internal representations, while policy / value models usually share same state input. However, shared networks are more challenging due to gradients from different losses varying in direction and scale, requiring careful tuning. Thus, stabilizing the training of shared networks is also one of our contributions, in the sense of RL algorithm. Additionally, inferencing with two separate LLMs would almost double the computational cost.
- **Inference method** Unlike previous works, we mainly use MCTS for training rather than inference. For inference, we propose the faster SBS modified from MCTS.

[r1] Sutton & Barto's Reinforcement Learning: An Introduction

[r2] The UC Berkeley course: Deep Reinforcement Learning (CS285)

> G2: Explanation between maj@n and SBS(1, n)

We use an example to illustrate the difference between maj@n and SBS(1, n) with $n=5$ and max depth=3.
- **maj@5** The final results are 5 solution paths $(a_1^j,a_2^j,a_3^j)^{j=1:5}$, where $a_t^j\ne a_t^k$ if $j\ne k$. E.g., the recurrent relation is that $a_3^j$ is sampled based on $a_{1:2}^j$ for each $j$. The sampling is from 5 different partial solution paths, and all previous candidates must be maintained.
- **SBS(1, 5)** The final result is a top-1 solution path $(a_1,a_2,a_3)$, where $a_3$ is best action selected from $a_3^{1:5}$. Note that the recurrent relation is that all $a_3^{1:5}$ are sampled based on $a_{1:2}$, where the sampling is from only 1 best partial solution path. This step-level streaming manner is more friendly to production.

Thus, in the sense of diversity sampling, maj@n has more advantage than SBS(1, n). However, SBS with the value model can reverse this advantage. For the response purpose, we also include the comprehensive comparison in `Table 6 of PDF file`.

> G3: W9 and Limitations for **Reviewer 15WC**

Due to characters limitation, we apologize for putting the response in general rebuttal.

**W9**: Q-value in Fig 5 sampled from? For full or partial steps? Missing legend?

With the Q-value update formula, MCTS assumes it is finally converged. Then we used the converged Q-values of all nodes for plotting Fig 5.

In `Lines 279-280`, we mentioned the left value plot is only for intermediate steps on trainset (the terminal value is final reward, no need to estimate). In `Lines 292-293`, we mentioned the right value plot is for both intermediate steps and terminal steps on testset.

We carefully reviewed the right figure and confirmed that it indeed contains only two curves and two legends.

**L1**: False positive / negative issues

We acknowledge that there are possible false positive / negative intermediate steps, especially for the proof (text analysis in REACT). We will include it in future version.

For calculation issue, the false cases can mostly be solved.
- Practically, in the `Solution Filtering Algorithm 3 of Appendix C.2`, python code can verify the numerical errors. Pure text reasoning (CoT in Math-Shepherd, OmegaPRM) may encounter the recent famous error $9.9<9.11$.
- Theorectially, our value estimation is based on MCTS. The Q update of an intermediate step in MCTS can achieve better estimation by considering all its children nodes in a complex weighted form.

**L2**: Not always applicable to many tasks

Thank you for your insightful comment. In `Lines 516-520`, we partially mentioned this issue. To ensure greater clarity, we will add it to **Limitations** in future version.

---

### Decision · Program_Chairs · 2024-09-25

**Decision:**

Accept (poster)

**Comment:**

The paper tackles solving mathematical reasoning with LLMs. The key idea is to leverage Monte Carlo Tree Search (MCTS) to sample high-quality process-supervised data, which currently existing works primarily rely on expensive human annotation or GPT-4. Moreover, they propose an efficient inference strategy called step-level beam search, which leverage value models trained from the generated process supervised data. Altogether demonstrates significant improvements on various datasets, showing comparable or even superior results to existing state-of-the-art methods without relying on external process annotations.

The reviewers all gave borderline scores. Based on my assessment, most of their raised weaknesses have been addressed by the author during rebuttal. However very few engagement during and post rebuttal happened, despite multiple reminders from the AC. One of the reviewers giving a rating of 4 agreed that the paper was "very close to the threshold" and "a valuable study in this area" if the authors can properly address their concerns, which, to me, they did.

I therefore recommend accepting despite the slightly low average score.